# Hierarchy and control of ageing-related methylation networks

**Gergely Palla**[1,2], **Péter Pollner**[1,2]*, **Judit Börcsök**[3], **András Major**[4], **Béla Molnár**[5], **István Csabai**[4]

1 Health Services Management Training Centre, Semmelweis University, Budapest, Hungary, 2 MTA-ELTE Statistical and Biological Physics Research Group, Dept. of Biological Physics, Eötvös University, Budapest, Hungary, 3 Danish Cancer Society Research Center, Copenhagen, Denmark, 4 Dept. of Physics of Complex Systems, ELTE Eötvös University, Budapest, Hungary, 5 Molecular Medicine Research Group, Hungarian Academy of Sciences, Budapest, Hungary

* peter.pollner@emk.semmelweis.hu

**Data Availability Statement:** The complete methylation profiles and the metadata we use is publicly available in NCBI's Gene Expression Omnibus (GEO) under accession number GSE40279.

## Abstract

DNA methylation provides one of the most widely studied biomarkers of ageing. Since the methylation of CpG dinucleotides function as switches in cellular mechanisms, it is plausible to assume that by proper adjustment of these switches age may be tuned. Though, adjusting hundreds of CpG methylation levels coherently may never be feasible and changing just a few positions may lead to biologically unstable state.

A prominent example of methylation-based age estimators is provided by Horvath's clock, based on 353 CpG dinucleotides, showing a high correlation (not necessarily causation) with chronological age across multiple tissue types. On this small subset of CpG dinucleotides we demonstrate how the adjustment of one methylation level leads to a cascade of changes at other sites. Among the studied subset, we locate the most important CpGs (and related genes) that may have a large influence on the rest of the sub-system. According to our analysis, the structure of this network is way more hierarchical compared to what one would expect based on ensembles of uncorrelated connections. Therefore, only a handful of CpGs is enough to modify the system towards a desired state.

When propagation of the change over the network is taken into account, the resulting modification in the predicted age can be significantly larger compared to the effect of isolated CpG perturbations. By adjusting the most influential single CpG site and following the propagation of methylation level changes we can reach up to 5.74 years in virtual age reduction, significantly larger than without taking into account of the network control. Extending our approach to the whole methylation network may identify key nodes that have controller role in the ageing process.

## Author summary

Aging affects all living organisms. In humans, the chronological age correlates with the methylation level of some locations of the DNA. Here we extract an interaction network between these ageing related sites, which shows signs of hierarchical organisation. In

**Funding:** The research was partially supported by the Velux Foundation 00018310 (JB) and by the Hungarian National Research, Development and Innovation Office K128780 (PG, PP) and by the Hungarian National Research, Development and Innovation Office NVKP\_16-1-2016-0004 (JB, AM, BM, ICs) and by the Research Excellence Programme of the Ministry for Innovation and Technology in Hungary, within the framework of the Digital Biomarker thematic programme of the Semmelweis University (GP, PP) and the NRDI Office within the framework of the Artificial Intelligence National Laboratory Program (ICs). The funders had no role in study design, data collection and analysis, decision to publish, or preparation of the manuscript.

**Competing interests:** The authors have declared that no competing interests exist.

addition, modifications in the methylation of single sites of the DNA can impose cascades of changes at other sites over this network. Based on "gedanken-experiments" in a small subset of CpG sites we show that by tuning appropriately selected methylation levels the estimated biological age can be changed. When modifying the most influential locations, the resulting cascades of changes can set back the estimated biological age by more than 5 years. Our study also shows that compared to single site methylation perturbations, the propagation of the change over the interaction network leads to methylation change profiles which are more aligned with the natural direction of ageing in a high dimensional representation of the methylation levels.

## Introduction

An ancient desire of humanity is to understand, slow, or even halt and reverse ageing. In related studies it was soon realised that certain biomarkers can rather precisely predict the functional capability of tissues, organs and even patients [1, 2]. In addition, age-related biomarkers enable the introduction of biological age [3, 4], bringing additive information in the risk assessments for age-related conditions on top of chronological age. Individuals of the same chronological age can still show great heterogeneity in the tissue and organismal functions, and thus, could possess different risks for age-associated diseases as judged from their biological ages. However, the predictive value of biological age estimators is usually decreasing at old age due to the increased biological heterogeneity among elderly individuals [3].

Probably the most promising age-predictive biomarkers are the ones based on DNA-methylation [5–7], which can be used for basically any source of DNA from sorted cells through tissues to organs, and can predict the biological age across the whole life span from prenatal tissues to tissues obtained from centenarians [5]. DNA methylation-related markers are also important in endocrinology [8], cell biology [9], biodemography [10], lifestyle factors [11], and medicine [12].

The research of DNA methylation dates back to the 1960s, when it was first observed that the methylation level of the CpG dinucleotides in the DNA is changing genome-wide with the chronological age [13, 14]. Later on, thanks to the developments in methylation array technologies, specific CpG dinucleotides were located in the genome, based on which the age of the DNA source (e.g., a tissue, an organ, or a person) can be estimated [15–22]. In fact, age related change in DNA methylome turns out to be quite common, where up to 15—30% of all CpG sites is displaying changes of certain types related to ageing. The involved change can be random fashion due to epigenomic drift [23] directional, or show increased variability with age [24]. The research on DNA-methylation related biomarkers has been a huge success [25], with still some important challenges remaining, such as the dissection of the regulators and drivers of age-related changes in single-cell, tissue- and disease-specific models, the analysis of further epigenomic marks, the implementation of longitudinal and diverse population studies, and the exploration of non-human models [26].

In general, when the goal is to estimate the age based on the methylome, the usual framework couples a set of CpGs with a mathematical algorithm, where the observed methylation levels of the CpG dinucleotides are combined in some way to yield the estimated age in years [25]. The obtained estimated age is referred to as DNAm age, or epigenetic age, which is highly correlated with the chronological age, but is also effected by other biological factors such as the health status as well. The above mentioned DNA methylation-based age estimators are usually built using supervised machine learning techniques such as penalized regression models,

which automatically select the most informative CpGs for the age estimation [25]. The first DNA methylation-based age estimators in the scientific literature were concentrating on a single tissue [27, 28], and therefore, were tailor-made for just one type of DNA source, leading to biased estimates for other tissues. However, the construction of multi-tissue DNA methylation-based age estimators is non-trivial, due to the significant differences between the DNA methylation patterns among different tissues [29, 30], the specific ways in which the DNA methylation patterns change with age across the different cell types [17, 31], the fact that distinct biological processes drive the observed age-related hypermethylation and hypomethylation, and that baseline DNA methylation state is strongly driven by genetics being highly CpG density dependent [32].

Nevertheless, after the discovery of CpGs that show age related changes in a diverse range of tissues [20], Steve Horvath proposed the first multi-tissue DNA methylation-based age estimator, which is often referred to as Horvath's clock [5]. This estimator is based on elastic net regression, and was trained and validated using 8,000 microarray samples from over 30 different tissue and cell types collected from patients of age ranging from children to adult, selecting altogether $N = 353$ CpGs from the overall 27k CpG dinucleotides in the data. According to the results of the elastic net regression method, by taking the measured methylation levels $m_i(q)$ for patient $q$ at the $i$-th CpG dinucleotide, the estimated age $a(q)$ can be given as

$$a(q) = \begin{cases} (a_t + 1)\sum_{i=1}^{N} H_i m_i(q) + a_t & \text{if } \sum_{i=1}^{N} H_i m_i(q) \geq 0, \\ \exp\left(\sum_{i=1}^{N} H_i m_i(q) + \ln(a_t + 1)\right) - 1 & \text{if } \sum_{i=1}^{N} H_i m_i(q) < 0, \end{cases} \quad (1)$$

where the adult age threshold $a_t$ is equal to $a_t = 20$, and $H_i$ is corresponding to the coefficient of the $i$-th CpG dinucleotide. Over the years, Horvath's clock turned out to be a very successful age estimator, providing accurate results using various DNA sources across the entire human lifespan [25], although some caveats still remain [33, 34]. In a recent study, together with similar clocks derived by Hannum et. al [28], Levine et al. [35] and Lu et al. [36], Horvath's clock was used to measure the impact of a protocol intended to regenerate the thymus, where the mean epigenetic age was 1.5 years less than baseline after 1 year of treatment [37].

The estimation of age based on DNA methylation across multiple tissues is indeed a complex problem. E.g., in case of Horvath's clock, for 193 of the included CpGs the methylation state is positively correlated with age, whereas for the remaining 160 CpGs we can observe a negative correlation [5]. Furthermore, when considered individually, the methylation state for most of the CpGs is only weakly correlated with age, e.g., the strongest and most robust individual CpG pan-tissue changes from the ELOVL2 locus [38, 39] were not included in the clock. Thus, these were selected not for their individual strength, but rather to their power to work collectively to parsimoniously capture ageing over the life-course [25]. This also means that we cannot really point out any of these CpGs as being more important than others for measuring the molecular age. These properties are in full consistency with the way the elastic net regression method is selecting among the feature variables by penalising the coefficients both in quadratic and absolute forms. In this approach by combining the penalty of the summed absolute value of the coefficients from Lasso regression (that is known to turn the coefficients of unimportant variables to zero) with the quadratic penalty of the coefficients from Ridge regression we obtain a convex loss function with a unique minimum, making the selection more stable.

When considering the modifications to the methylation profile during the life-course, the known ageing effects are leading to coordinated changes across the entire DNA methylome, including those driven by cell-type specific epigenomics, where changes in cell proportion will led to variation (including the age-related myeloid skew [40], T cell exhaustion [41]), polycomb target hypermethylation [20], bivalent domain hypermethylation [42], etc. Such systemic effects can be seen as networks of age-related change, where the methylation level change of any CpG is accompanied by changes in the levels of related other CpGs as well.

In the present paper we examine this network of connections between the CpG dinucleotides of Horvath's clock. The tool we use to reveal the links is given by Lasso cross-validation (Lasso-CV) regression, which is a simple and robust approach to extract the most relevant inter relations between a given outcome (or response variable) and a larger set of regressors (or feature variables). Due to the complex nature of the problem, instead of focusing solely on the revealed pairwise interactions, we gather the obtained connections (links) into a methylation network, and analyse the properties of the system using techniques from complex network theory.

The network approach for studying the structure and dynamics of complex systems has become ubiquitous over the last two decades, and studies of networks ranging from gene interactions to the level of the society have shown that the statistical analysis of the underlying graph structure can highlight non-trivial properties and reveal previously unseen relations [43–46]. In the present work our network analysis is focusing on the hierarchical and control properties of the web of connections between the CpG dinucleotides.

Signs of hierarchical organization were observed in complex networks of diverse types [47], ranging from transcriptional regulatory networks within cells [48], through flocks of various animal species [49, 50], to the level of on-line news content [51], scientific journals [52], social interactions [53–55], ecological systems [56], and evolution [57]. In a hierarchical network, nodes close to the top of the hierarchy usually have a larger influence compared to nodes at the bottom levels. One of the related questions we address in this paper is whether we can detect signs of hierarchy in case of the methylation network too, and if so, which CpGs are on the top of the hierarchy?

Besides the hierarchical properties, another important aspect we investigate is given by the control properties of the network. The control theory of networks is based on the framework of structural controllability of linear dynamical systems [58], exploiting the connections between graph combinatorics and linear algebra. The importance of the nodes from the point of view of control can be characterised by the control centrality [59], corresponding to the number of nodes we can drive by controlling the given node using an external signal. By combining control centrality with the results from the hierarchy analysis, we can locate the CpGs having the largest influence on the rest of the system, which may also play a crucial role in the process of ageing.

In close relation to the above, we also introduce a simple framework, in which we can examine the effect of perturbing the methylation level of CpGs on the DNAm age obtained according to Horvath's clock. The basic idea is initiating a small change on the methylation of one CpG, and then propagate the effect over the methylation network according to the regression coefficients defining the link weights. This provides a minimal model for tracking the changes in the methylation profile, in which the complex interrelations between the CpGs are taken into account. In our view, treating the set of ageing-related CpGs as a complex, interacting system can provide more realistic change profiles in DNA methylation compared to isolated individual shifts in the methylation of single CpGs. Nevertheless, there can still be

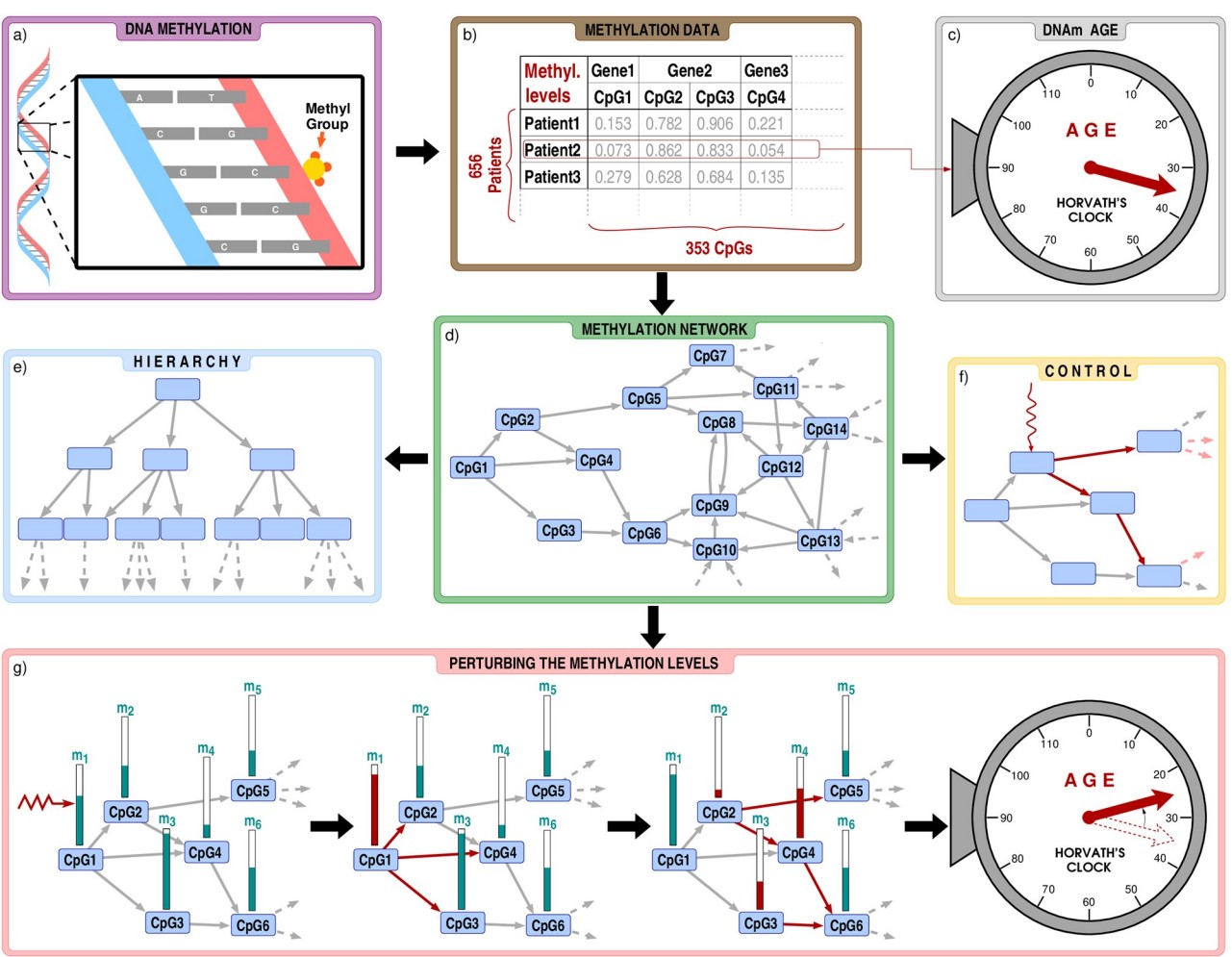

**Fig 1. Flow chart of our analysis.** a) Our study is based on cytosine methylation, a phenomenon where a methyl group is attached to a CpG dinucleotide in the DNA. b) We focus on the methylation level of the 353 CpG dinucleotides appearing in Horvath's clock, using the data from Ref. [28], listing altogether 656 patients. c) By plugging in the methylation levels of a given patient into Horvath's clock, we obtain the DNAm age, which is in strong correlation with the chronological age, but is also affected by e.g., the health status. d) Using Lasso-regression, we construct a methylation network between the CpG dinucleotides. In order to seek for key influential nodes in the system we analyse the hierarchical (panel e) and control properties (panel f) of the network. g) In addition, we also investigate how would the change of the methylation levels affect the estimated age when the perturbations are transmitted over the methylation network.

differences between the obtained shifts in the DNAm age depending on which CpG was chosen for the initiating perturbation, and naturally, one of the most interesting questions is for which initiating CpG do the methylation changes accumulate in such a way that the resulting shift in the DNAm age is maximal. A flow chart summarising the investigations carried in the paper are given in Fig 1.

There is an important caveat to keep in mind when interpreting our results is that we demonstrate the hierarchy, control and perturbation of the methylation network only on a small, somewhat 'artificially' isolated subset of the orders of magnitude large set of all CpG dinucleotides of the human genome. To uncover biologically relevant factors the presented network control analysis should be extended to more sites and possible to other mechanisms beyond DNA methylation.

## Results

### Methylation network based on Lasso-CV regression

We constructed the methylation network between the 353 CpG dinucleotides of Horvath's clock by repeatedly applying Lasso-CV regression for predicting the methylation level of one of the CpG dinucleotides based on the methylation level of the 352 other positions in the data set from Ref. [28] (more details on the data set are given in Methods). In this approach, the methylation level $m_j$ of CpG dinucleotide $j$ can be modelled as

$$m_j = \sum_{\substack{i=1 \\ i \neq j}}^{N} \beta_{ji} m_i + \beta_0,$$

(2)

where $\beta_{ji}$ denote the Lasso regression coefficients (with $\beta_0$ corresponding to the intercept). According to this model, the weight (or strength) $w_{ij}$ of a directed link from CpG dinucleotide $i$ pointing to CpG dinucleotide $j$ is given by $w_{ij} \equiv |\beta_{ji}|$.

The advantage of Lasso regression is that due to a factor $\alpha$ penalizing the $L^1$-norm of the vector of the regression coefficients, a fraction of the coefficients becomes exactly zero, resulting in a model that is more easy to interpret. For choosing the right value of $\alpha$ at each CpG dinucleotide, we used the well known method of cross-validation, more details on the applied Lasso-CV method are given in Methods. In order to make the resulting network sparse, we further applied a weight threshold $w^*$ on the link weights $w_{ij}$, and neglected connections where $w_{ij}$ was smaller than $w^*$. The optimal value of $w^*$ was set according to a general link weight thresholding method for biological networks based on the concept of efficiency [60], more details of this approach are given in Methods.

### Hierarchical properties according to $m$-reach

In order to study the hierarchical properties of the methylation network we use the concept of the Global Reaching Centrality [61], corresponding to a hierarchy measure that turned out to be successful in quantifying the extent of hierarchy in the structure of directed networks [52, 55, 61]. The basic idea behind this approach is that in a hierarchical network it should be easy for leaders (at the top of the hierarchy) to send orders or instructions to other nodes over relatively short paths, whereas this is not the case for nodes at the bottom of the hierarchy. Therefore, by comparing the fraction of nodes reachable in at most $m$ steps from a given node (called as the $m$-reach of the node) we can judge which nodes should be positioned high in the hierarchy, and which nodes are supposed to be at the lower levels.

In addition, based on the inhomogeneity of the $m$-reach distribution we can also quantify the extent of hierarchy in the organisation of the network. Technically this is done by using the Global Reaching Centrality denoted by GRC($m$), corresponding to the average difference between the individual $m$-reach of the nodes and the maximal $m$-reach value in the network (more details are given in Methods). In Fig 2 we show the GRC($m$) at $m$-values ranging from $m = 2$ to $m = 5$, where the value observed in the methylation network is compared to the empirical probability density of the GRC($m$) values measured in random directed networks with the same degree sequence. According to the results, the GRC($m$) distribution for the random network ensemble shows a bell shaped curve with a well defined average for all $m$ parameter values. The GRC($m$) measured for the original methylation network is way larger than this average, and the relative difference (measured in the units of the standard deviation $\sigma$) seems to be increasing with $m$. This means that the organisation of the network is far more

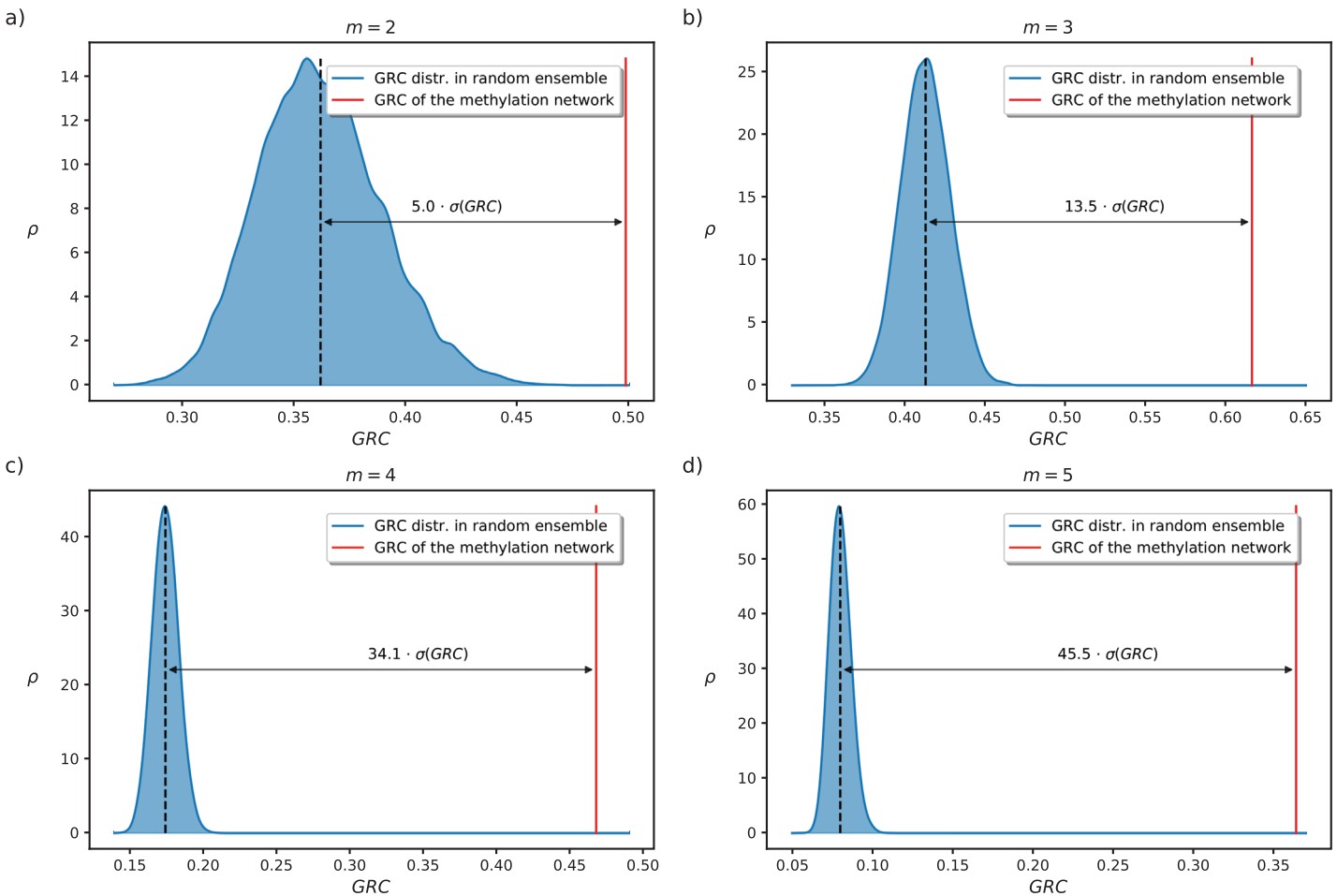

**Fig 2. Hierarchy of the methylation network.** We show the GRC($m$) measured for the network (red) together with probability density $\rho$(GRC) of the corresponding values in a link randomised ensemble of 20,000 networks (blue) at $m = 2$ (panel a), $m = 3$ (panel b), $m = 4$ (panel c), and $m = 5$ (panel d).

hierarchical compared to what we would expect in a random network with the same degree distribution.

Based on the $m$-reach, following the method suggested in Ref. [61], we can also apply a hierarchical layout to the network, as shown In Fig 3. Since higher $m$-reach means larger influence and therefore, higher position in the hierarchy, we can sort the nodes into levels according to their $m$-reach. In parallel, nodes in the same level are intuitively expected to have a similar influence, thus, it is quite natural to define the levels by grouping the nodes in such a way that the difference between the $m$-reach for any two members of a given level is below a certain threshold. The considerable number of links pointing upwards in the hierarchy highlights that the network is hierarchical in a non-trivial manner, i.e., its structure is far from a tree or a directed acyclic graph. However, in the mean time, the $m$-reach of the top nodes is still far larger compared to the average.

## Control centrality analysis

The control properties of the network can also be of high interest. Here we study the relative control centrality $c(i)$, which for any node $i$ is given by the maximal fraction of nodes we can

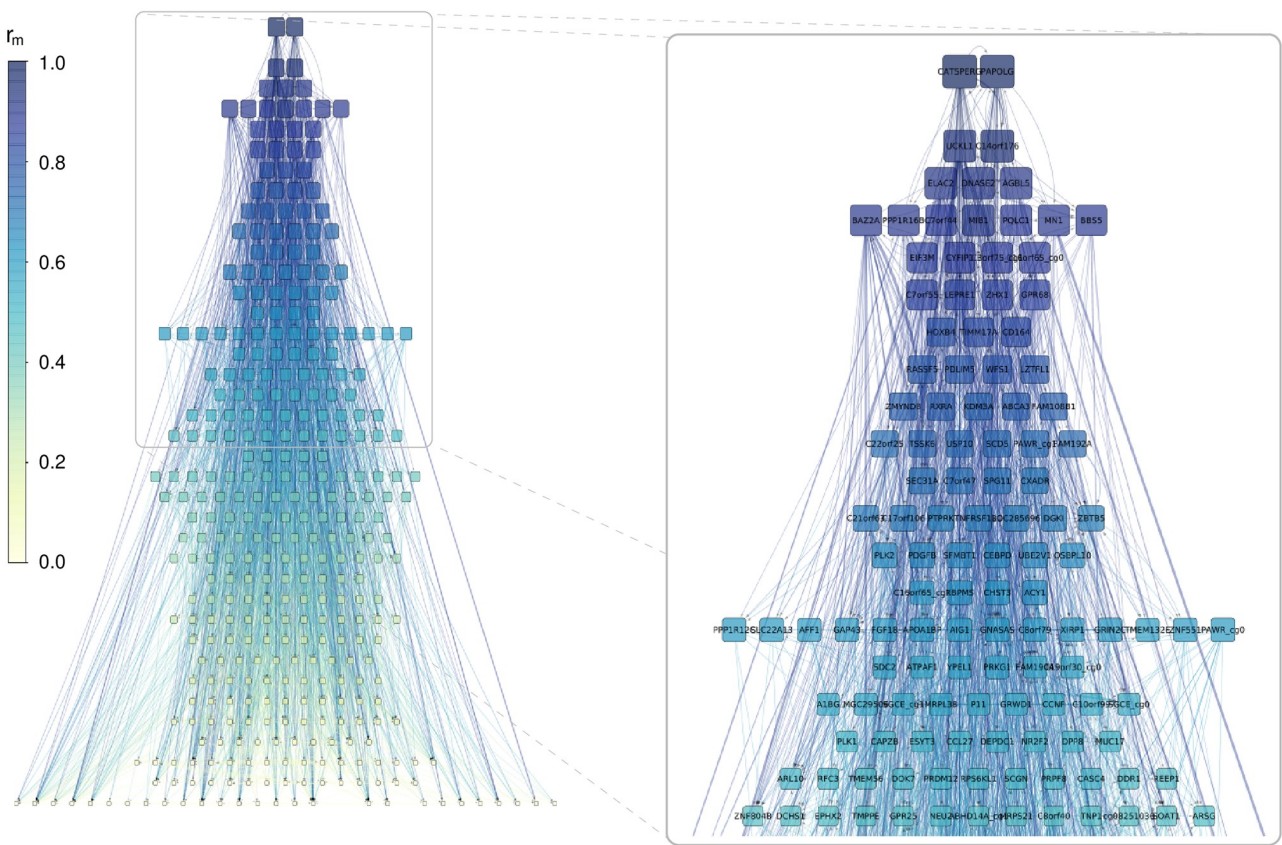

**Fig 3. Hierarchical layout of the network.** The nodes are sorted into the levels according to their $m$-reach at $m = 3$, where the levels are defined such that the difference between the $m$-reach for any pair of nodes in the same level is less then one-tenth of the standard deviation of the $m$-reach distribution. The shade and size indicates the $m$-reach, and the panel on the right shows the top of the hierarchy zoomed in. The $m$-reach of the two nodes at the top layer (CATSPERG and PAPOLG) is close to $r_{m=3} = 0.95$, thus, they can reach about 95% of the other nodes at most in 3 steps.

drive by controlling $i$. (More details are given in Methods). In Fig 4 we provide a scatter plot of the $m$-reach at $m = 3$ and $c(i)$. The top panel of the figure displays the probability density $\rho(c)$ for $c$, consisting of basically two narrow peaks for the methylation network at the optimal link weight threshold, shown by blue colour. In order to allow a finer distinction between the control abilities of the nodes, we calculated the control centrality for a larger number of networks obtained at link weight thresholds $w^*$ different from the optimal one, ranging in an interval between $w^* = 0.04$ and $w^* = 0.1$. By taking the average of $c(i)$ for a given node $i$ over these networks we obtain a quantity characterising its potential for driving other nodes in not a single instance of the methylation network, but rather, across the entire collection of networks. According to the top panel of Fig 4, the probability density $\rho(c)$ for the averaged control centrality (orange colour) has a far more complex structure compared to the $\rho(c)$ of the methylation network at the optimal $w^*$, and therefore, allows a finer ranking between the nodes.

Interestingly, the nodes that are important from the point of view of control, are usually also in high position in the hierarchy, as indicated by the scatter plot in the main panel of Fig 4. In case of the network at the optimal weight threshold $w^*$, the nodes with zero control centrality have also zero $m$-reach (set of blue points at the origin), whereas $r_m$ for nodes with high $c$ value can range practically between zero and one (set of blue points forming a vertical line on the right). In case of the $c$ averaged over the networks obtained at different weight thresholds

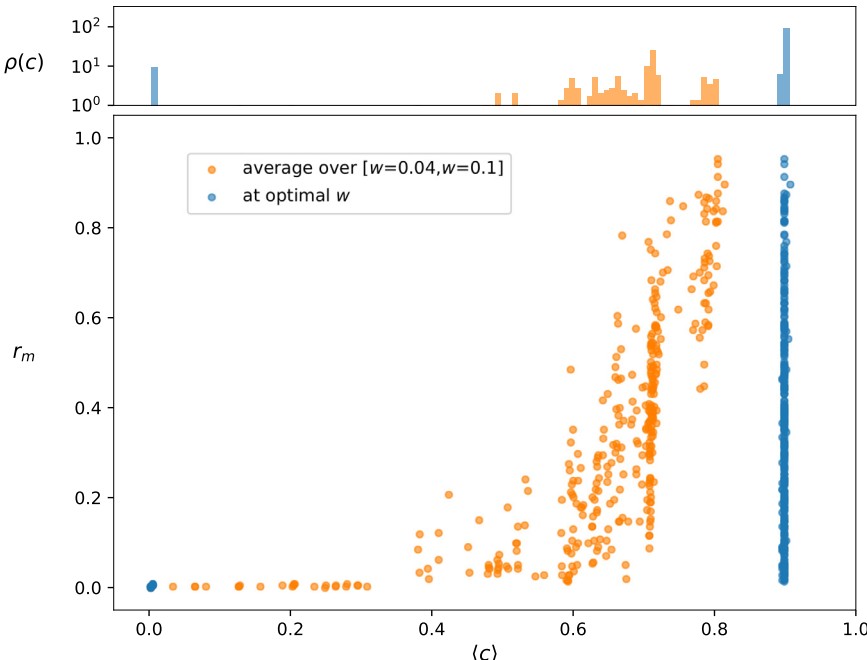

**Fig 4. Control centrality and reach.** The main panel shows the reaching centrality $r_m$ at $m = 3$ as a function of the relative control centrality $c$. Each symbol in the plot is corresponding to an individual CpG dinucleotide (node in the methylation network). In blue we show the results for the methylation network at the optimal weight threshold $w^*$, whereas in case of the orange symbols $C(i)$ was averaged for the individual nodes over 60 different networks obtained by changing the $w^*$ parameter in the $[w^* = 0.04, w^* = 0.1]$ interval. The Pearson correlation coefficient between $\langle c \rangle$ and $r_m$ is 0.41 for the optimal network, and 0.70 in case of the averaging scenario. The top panel displays the density of the normalised control centrality $c$ for the two cases.

(orange colour), the bulk of the point cloud shows an increasing tendency, whereas we can also see a narrow line of points with zero reach in the low $c$ regime.

## Modifying the predicted age by perturbing the methylation network

A natural question arising during the analysis of the methylation network is how do the perturbations of the methylation levels affect the estimated age, and which are the CpG dinucleotides where the observed sensitivity in the predicted age is maximal for small perturbations? In the simplest scenario we can consider perturbing the methylation levels of the individual CpG dinucleotides one by one, without taking into account any possible propagation of such effect over the methylation network. Since the vast majority of the patients in the used data are older than the adult age threshold $a_t$ appearing in Eq (1), for simplicity let us assume that the estimated age is calculated according to the first equation in (1), which is linear in both $H_i$ and $m_i$. Thus, if we can change one and only one $m_i$ value, then the largest effect for a unite change in $m_i$ is expected for the CpG dinucleotide $i$ where $\left| \frac{da}{dm_i} \right| = (a_t + 1)|H_i|$ is maximal.

However, perturbing the methylation level of a single CpG is quite likely to induce changes in the methylation of further other CpGs as well. A living tissue is reacting to outside perturbations, and these reactions plausibly affect the methylation of the related other CpGs. Since the links of the methylation network encapsulate the most relevant linear connections between the methylation levels, using these we can examine the expected propagation of the change in the overall methylation pattern. To keep our framework simple, we assume that

initially the methylation level of a single CpG dinucleotide is changed as $m_i \rightarrow m_i + \Delta m_i$ due to some outside perturbation, and then this change triggers further modifications in the methylation of other CpG dinucleotides as well. In general, we may take into account chains of interactions up to some maximal length $\ell_{max}$. When $\ell_{max} = 0$, we actually have only single node perturbation, $\ell_{max} = 1$ is corresponding to taking into account the first neighbour interactions, $\ell_{max} = 2$ includes also the 2$^{nd}$ neighbour interactions, etc.

According to Eq (2), a $\Delta m_i$ perturbation of the methylation level of CpG dinucleotide $i$ is inducing a change in the methylation level of node $j$ given by $\Delta m_j = \beta_{ji}\Delta m_i$. Therefore, if we take into first neighbour interactions ($\ell_{max} = 1$), for the derivative of the estimated age with respect to $m_i$ we can write

$$\frac{1}{a_t + 1}\frac{\mathrm{d}a}{\mathrm{d}m_i} = H_i + \sum_{j \in L_i^{out}} H_j \beta_{ji}, \tag{3}$$

where $L_i^{out}$ denotes the set of out-neighbours for $i$. When including longer chains of interactions in the network ($\ell_{max} > 1$), due to the linear nature of the problem, the only modification to the above is that we have to multiply the $\beta$ coefficients along the considered paths, yielding

$$\begin{aligned}
\frac{1}{a_t + 1}\frac{\mathrm{d}a}{\mathrm{d}m_i} = & \ H_i + \underbrace{\sum_j H_j \beta_{ji}}_{\text{1st neighs.}} + \underbrace{\sum_{jk} H_j \beta_{jk}\beta_{ki}}_{\text{2nd neighs.}} + \underbrace{\sum_{jks} H_j \beta_{js}\beta_{sk}\beta_{ki}}_{\text{3rd neighs.}} + \cdots = \\[2mm]
& H_i + \sum_j H_j \left( \beta_{ji} + \sum_k \beta_{jk}\beta_{ki} + \sum_{ks} \beta_{js}\beta_{sk}\beta_{ki} + \cdots \right) = \\[2mm]
& H_i + \sum_j H_j \underbrace{\sum_{u \in \mathcal{D}(i,j)} \prod_{q=i}^{\ell(u)+i} \beta_{q+1,q}}_{\text{effective } \beta_{ji}},
\end{aligned} \tag{4}$$

where $\mathcal{D}(i,j)$ denotes the set of allowed paths between $i$ and $j$, the length of a path $u \in \mathcal{D}(i,j)$ is given by $\ell(u)$, and the term $\prod_{q=i}^{\ell(u)+i} \beta_{q+1,q}$ is simply the product of the Lasso regression coefficients along $u$. The result has a very similar form compared to (3), and can be interpreted as that the longer chains of interactions introduce an effective $\beta_{ji}$, corresponding to the sum of the products of the original $\beta$ values along the considered paths between $i$ and $j$.

Let us now move from the derivative of the estimated age to the expected actual change in $a$, which is given by the product of the age derivative given in (4), and $\Delta m_i$ as

$$\Delta a = \frac{\mathrm{d}a}{\mathrm{d}m_i}\Delta m_i. \tag{5}$$

In order to keep the framework realistic, let us assume that $\Delta m_i$ is of the order of magnitude given by the deviations we see in the data, thus, we set $\Delta m_i \equiv 2\langle\sigma(m)\rangle$, where $\langle\sigma(m)\rangle$ denotes the average of the standard deviation of the methylation levels among the different CpGs.

We have calculated the derivative of the estimated age and the corresponding corresponding $\Delta a$ for each node in the methylation network using $\ell$ parameters between $\ell = 0$ and $\ell = 1$ (the distribution of $|\Delta a|$ is shown in Fig A in S1 Text). The average for $|\Delta a|$ was observed to be $\langle|\Delta a|\rangle = 0.40$ for the $\ell_{max} = 0$ case, and $\langle|\Delta a|\rangle \simeq 0.63$ for the $\ell_{max} = 4$ scenario, meaning that the perturbation of the individual methylation levels has a larger effect on the estimated age if the chains of interactions between the CpGs are taken into account.

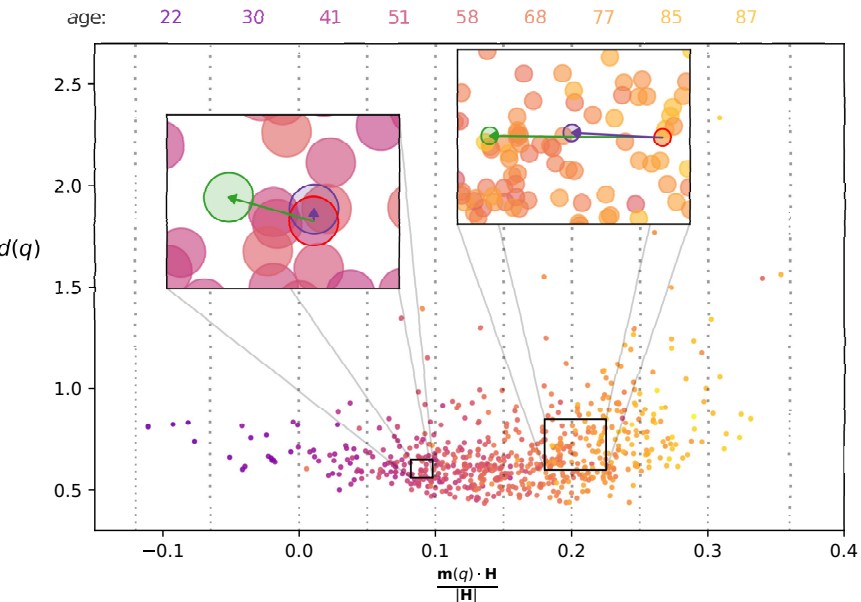

**Fig 5. The point cloud of patients in the space of methylation coordinates.** The horizontal axis is showing the projection of the patient vectors $m(q)$ onto $H$ (corresponding to a vector composed of the coefficients of Horvath's clock), whereas the vertical axis is showing the distance $d(q)$ from the center of mass in the hyper plane perpendicular to $H$. The colours represent the chronological age, with dark, colder shades indicating younger patients, and bright, warm shades signalling older patients. At the top of the figure we list the average age for the patients in the bins indicated by the dotted vertical lines. The left inset shows the displacement due to perturbing the methylation level of *GAP43*, with the purple node corresponding to the $\ell_{max} = 0$ case, and the green node indicating the $\ell_{max} = 4$ case. The right inset is showing the analogous results for perturbing the methylation level of *SCGN*.

Besides leading to a larger effect in the change of the estimated age, the scenario where we propagate the perturbation of the methylation levels over the network seems to induce change patterns that are more aligned with plausible changes in the methylation profiles due to ageing. To illustrate this, we define a high dimensional representation of the methylation profiles given by a Euclidean space where each dimension is corresponding to a CpG dinucleotide, and the methylation levels define the coordinates according to the corresponding axes. In our study the complete 'methylation space' is narrowed down to 353 dimensions, corresponding to the CpGs in Horvath's clock selected by elastic net regression, and the patients in the data set form a cloud of 656 points in the space.

Let us denote the vector pointing from the origin to patient $q$ as $\mathbf{m}(q)$, where the component $i$ of $\mathbf{m}(q)$ is simply given by $m_i(q)$. We can also define a vector $\mathbf{H}$ for which the component $i$ is equal to the coefficient $H_i$ appearing in (1). This way, the estimated age $a(q)$ for patients above the adult age threshold (top line in Eq (1)) can be also written as $a(q) = (a_t + 1)\mathbf{m}(q) \cdot \mathbf{H} + a_t$, where $\mathbf{m}(q) \cdot \mathbf{H}$ is corresponding to the scalar product (inner product) of the corresponding vectors. In Fig 5 we show a 2 dimensional projection of the point cloud of the patients, where the horizontal axis is corresponding to the component of $\mathbf{m}(q)$ pointing in the direction of $\mathbf{H}$ (given by $\frac{\mathbf{m}(q) \cdot \mathbf{H}}{|\mathbf{H}|}$), and the vertical axis is displaying the distance $d(q)$ from the centre of mass of the point cloud in the hyper-plane perpendicular to $\mathbf{H}$. The colouring of the nodes indicates the chronological age of the patients, and according to the figure, we can observe a clear correspondence between the chronological age and the estimated age based on Horvath's clock (which is proportional to the horizontal coordinate of the point).

The perturbations of the methylation profile we considered earlier can also be interpreted as vectors pointing in a certain direction in the methylation space, e.g., perturbing the methylation level of just a single CpG dinucleotide can be represented by a vector pointing in the direction of the corresponding base vector. For example, in the insets of Fig 5 we show the results when modifying the methylation level of *GAP43* (left inset) and of *SCGN* (right inset) by $2\langle\sigma(m)\rangle$ for two randomly chosen patients. The nodes marked by the red perimeter correspond to the chosen patients, and the purple nodes represent the new position when only the methylation level of *GAP43* or *SCGN* are modified (the $\ell_{max} = 0$ case). According to the figure, the displacement for *GAP43* (left inset) is rather small, and seems to be perpendicular to **H**, whereas in case of modifying the methylation level of *SCGN* (right inset), the displacement is larger, and is in good alignment with **H**. However, when we take into account the propagation of the perturbation over the methylation network, the length of the displacement for *GAP43* is increased, and it is also much more aligned with **H**, as indicated by the green node in the left inset. In parallel, for *SCGN* the displacement remains aligned with **H** when switching from $\ell_{max} = 0$ to $\ell_{max} = 4$, and its length is nearly doubled (green node in right inset). Similar effects can be observed for the other CpGs as well, for example the angle between the direction of change obtained at $\ell_{max} = 4$ and **H** is smaller for 296 out of the 353 CpG dinucleotides compared to the angle between **H** and the corresponding single CpG change direction. These examples suggest that taking into account the propagation of the perturbation over the methylation network, the induced change is more aligned with the 'natural direction of ageing'.

In addition, let us also examine the relation between $|\Delta a|$ and the previously analysed topological characteristics of the CpG dinucleotides. In Fig 6 we re-plot the hierarchy according to the $m$-reach (shown in Fig 3), however this time the size and colouring of the nodes is indicating the expected change in the estimated age, $|\Delta a|$ if we perturb the methylation level of the given node according to (4) and (5) at $\ell_{max} = 4$. The nodes with higher $|\Delta a|$ values tend to be placed higher in the hierarchy, however with also a strong variation among CpGs with similar $|\Delta a|$ values. E.g., *BAZ2A*, *UCKL1* and *AGBL5* from the top 5 nodes according to $|\Delta a|$ are very high up in the hierarchy, whereas *SCGN* (having the largest $|\Delta a|$ at $\ell_{max} = 4$ amongst all nodes) is at a relatively low level compared to them. Nevertheless, the position of a node in the hierarchy and its potential for inducing a large change in the estimated age are clearly interrelated.

Finally, in Fig 7 we show the scatter plot of $|\Delta a|$ at a maximal path length of $\ell_{max} = 4$ as a function of the $m$-reach $r_m$ and the average control centrality $\langle c \rangle$ of the node where the initial perturbation is applied. According to the figure, a moderate increasing tendency can be observed in the behaviour of $|\Delta a|$, which means that nodes with larger $m$-reach and/or higher control centrality are good candidates for achieving a notable change in the estimated age $a$ when perturbing the methylation level of the corresponding CpG dinucleotide.

## Extension of the analysis to further methylation networks

Although the results shown so far indicate that the methylation network of Horvath's clock is displaying interesting hierarchical and control properties, a question of key importance is how do these features generalise for different sets of CpGs, or for even the entire 450k set of CpGs in the input data. Relating to that we have examined methylation networks corresponding to both alternative epigenetic clocks and to sets of randomly chosen CpGs.

According to our results detailed in Section 2 in S1 Text, for the Skin-blood clock (consisting of $N = 391$ CpGs) and for Hannum's clock (having $N = 71$ CpGs) the networks obtained by applying the methodology presented here show a very similar behaviour compared to the

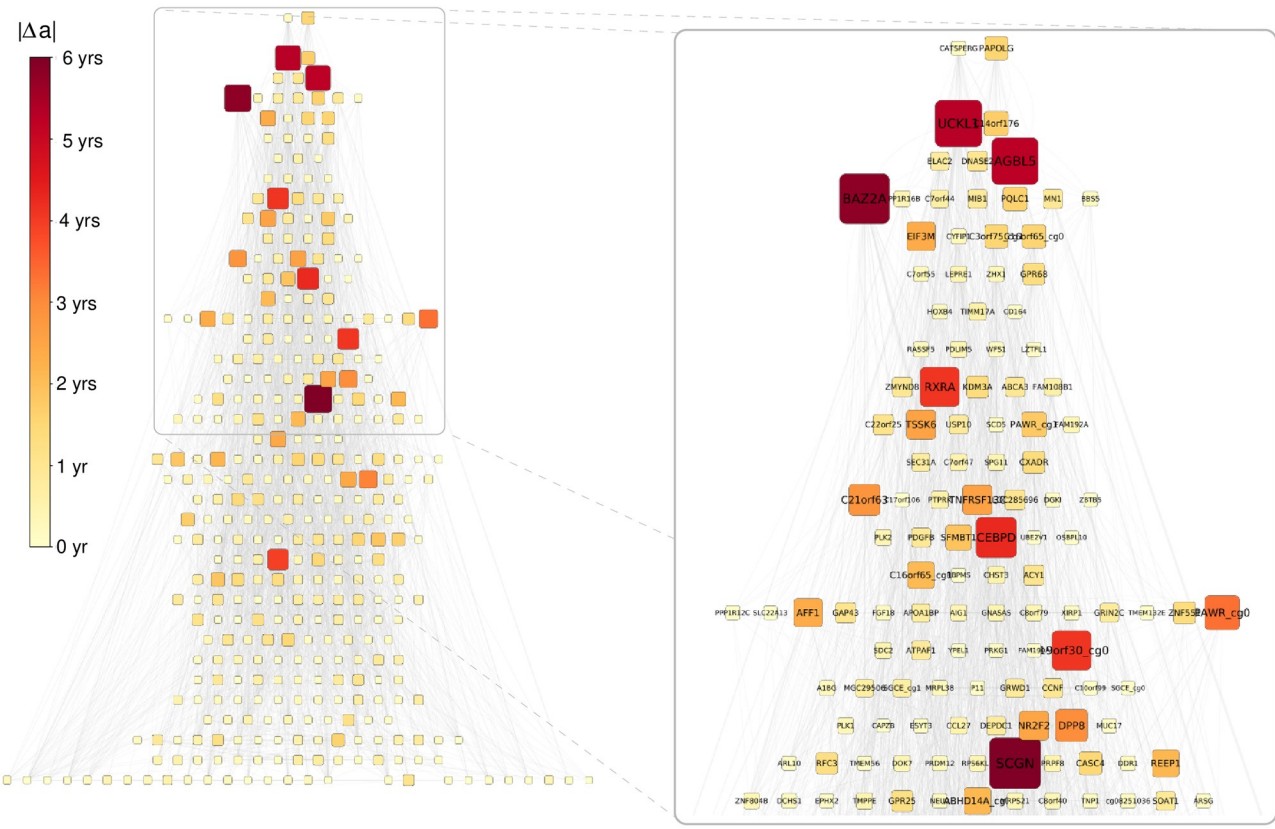

**Fig 6. Top levels of the hierarchy according to *m*-reach at *m* = 3.** The shading of the nodes indicates their estimated age reduction value |Δ*a*| (with darker shades corresponding to higher values).

methylation network of Horvath's clock. First, the GRC of these two networks is significantly higher than the average GRC in random graph ensembles with the same degree distributions as indicated by Figs B and C in S1 Text. Second, the control centrality of the nodes is in positive correlation with their hierarchy position determined by the *m*-reach in both systems, as shown by Figs D and E in S1 Text. Finally, when applying the framework of methylation level perturbations governed by Eqs (3)–(5), the expected change in the estimated age is usually higher than average if the perturbed node is chosen from the top part of the hierarchy with larger control centrality (Figs F and G in S1 Text), again, in a similar fashion to what we have demonstrated for the methylation network of Horvath's clock here.

In parallel with the Skin-Blood clock and Hannum's clock, we also examined the properties of methylation networks where random sets of CpGs (with equal size to the set of CpGs in Horvath's clock) were selected from the available 450k CpG in the data. Based on the results shown in Section 3 in S1 Text, these networks are also more hierarchical compared to their link randomised counterpart, although the difference between the average *GRC* value of the link randomised networks and the average *GRC* of the original networks is not as large (in the units of the standard deviation of the link randomised ensemble) as for the case of Horvath's clock. The control centrality is again in positive correlation with the node position in the hierarchy determined by the *m*-reach, as indicated by Fig H.

The above results suggest that methylation networks obtained in our approach (based on regularised regression between the methylation levels) show hierarchical properties in general,

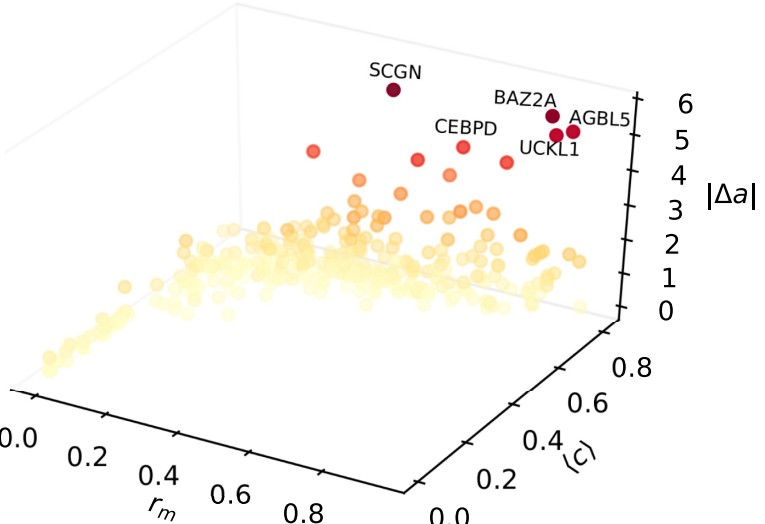

**Fig 7. Scatter plot of expected change in the estimated age as a function of the the *m*-reach and the control centrality.** The vertical axis is corresponding to $|\Delta a|$, calculated according to (5), where the initial perturbation $\Delta m_i$ on the methylation level is equal to $2\langle\sigma(m)\rangle$ for all $i$, and the age derivative for node $i$ is obtained from (4) at $\ell_{max} = 4$. On the $x$ axis we display the $m$-reach $r_m$ at $m = 3$, whereas the $y$ axis is corresponding to the average control centrality $\langle c \rangle$. The colouring of the symbols follows their vertical coordinate, with bright colours corresponding to low $|\Delta a|$ values, and darker shades representing a larger expected change in $a$.

that are positively correlated with the control centrality of the nodes. However, a remaining question of key interest is whether the nodes observed to be at the top of the hierarchy for small methylation networks could play a leading role also when considering e.g., the entire web of connections between the 450k CpGs in the data? Unfortunately, the scaling up of our approach to this size level is computationally unfeasible, nevertheless, we can still examine to what extent is the hierarchy position of certain nodes conserved when the set of CpGs defining the network is varied as follows. We have taken the top 10% of the nodes from the hierarchy seen for Horvath's clock (Fig 3), and "mixed" them together with randomly chosen CpGs from the 450k methylation array, forming networks of the same size (353 nodes) as in the original case of Horvath's clock. By applying the same methodology, we can examine the hierarchical properties of these mixed networks as well, with a special focus on the position of the nodes that were at the very top in case of the hierarchy corresponding to Horvath's clock.

In Fig 8a we show the hierarchical layout of 50 mixed networks plotted on top of each other, where the CpGs from the original network based on Horvath's clock are colored red. According to the figure, these nodes tend to be located at the higher levels in the mixed networks as well. In Fig 8b we display the analogous result when choosing the bottom 10% of the nodes from the hierarchy obtained for Horvath's clock to be mixed with random CpGs. The layouts show that in such a case the CpGs from Horvath's clock are located closer to the bottom of the hierarchy also for the mixed networks. These results indicate that the position of the nodes in the hierarchy is conserved to a considerable extent when the set of nodes defining the methylation network is varied.

Finally, we also examined the networks arising between the CpGs in Horvath's clock when applying our pipeline to an alternative methylation input data set. As described in Section 4 in S1 Text, by analysing the data studied by Lehne et al. in Ref. [62] we arrived to qualitatively very similar results as in case of the present data set. Namely, the extracted network was significantly more hierarchical compared to its configuration model ensemble counter parts (Fig I in

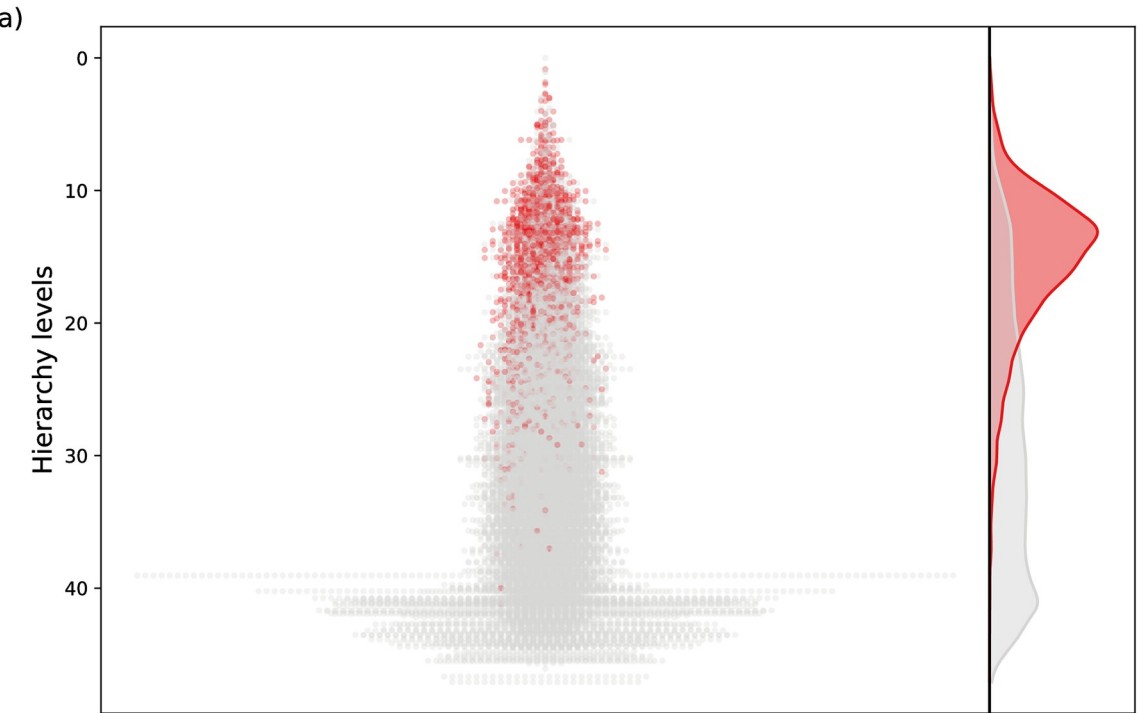

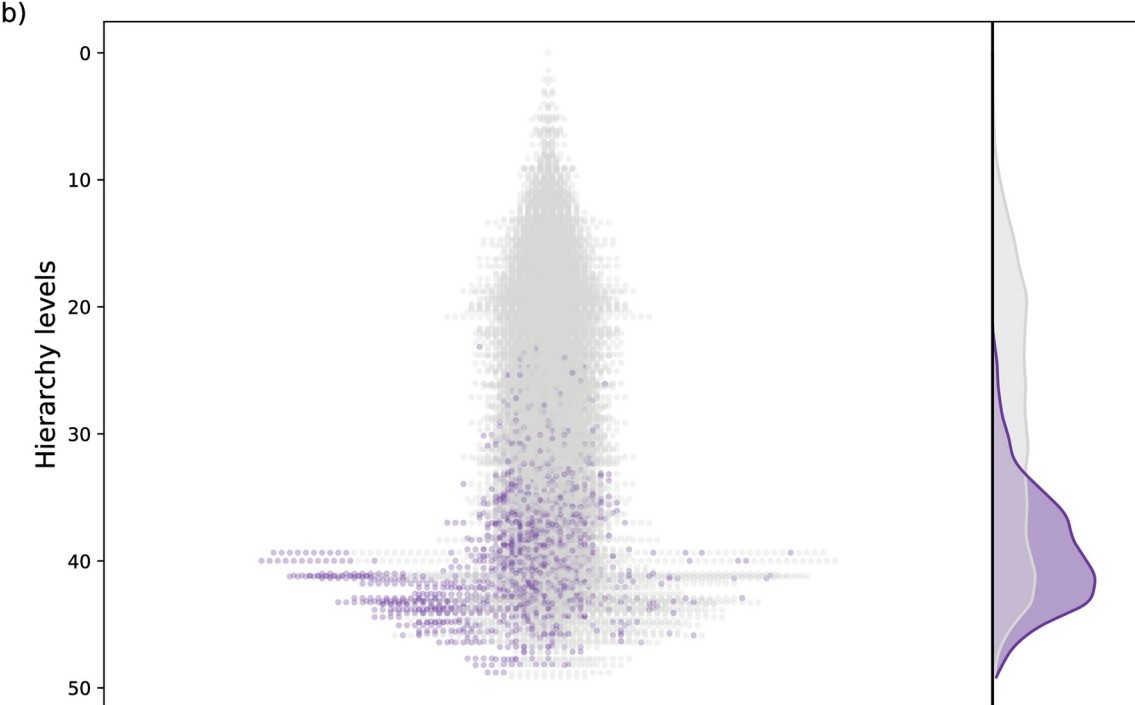

**Fig 8. Mixed hierarchies between randomly chosen CpGs where 10% of the nodes are from Horvath's clock.** a) When choosing the top 10% of the nodes from the hierarchy for Horvath's clock and mixing them with random CpGs, we obtain the hierarchies shown on the left, where 50 networks are projected on top of each other, with the nodes from Horvath's clock coloured red. On the right we display the density distribution across the hierarchy levels in grey for all nodes and in red for the nodes from solely Horvath's clock. b) The same as in panel a), but choosing the bottom 10% of the nodes from the hierarchy for Horvath's clock.

S1 Text), the control centrality and the $m$-reach were positively correlated (Fig J in S1 Text), and the best candidates for achieving a large change in the age predicted by Horvath's clock via perturbing the methylation level were the CpGs with a high position in the hierarchy and a large control centrality (Fig K in S1 Text). Furthermore, in spite of the significant difference between the age distribution of the two patient cohorts (as shown by e.g., Fig L in S1 Text), the hierarchies obtained for the Lehne et al. datset and the Hannum et al. data set show a relatively high similarity (Figs M and N and O in S1 Text), marked by e.g., a $C = 0.75$ Pearson correlation coefficient between the $m$-reach calculated in the two methylation networks. This indicates yet again the robustness of our analysis framework, that is capable of extracting consistent hierarchies between the CpGs based on diverse input methylation data sets.

## Discussion

According to our results, the methylation network between the 353 CpG dinucleotides of Horvath's clock is showing non-trivial hierarchical and control properties. Nodes with high standing in the hierarchy and/or a large control centrality are also likely to have more potential for inducing a large change in the estimated age when their methylation level is perturbed. In Table 1 we list the CpG dinucleotides (genes) that are in the top 20 according to either the change in the estimated age, the $m$-reach (position in the hierarchy), or the average control centrality. We can observe a high number of overlapping genes (11) among the top 20 genes of m-reach and the top 20 genes of control centrality supporting the correlation between m-reach and control centrality in the network. In the subsection *Biological role of top identified*

**Table 1. Top CpG dinucleotides according to our analysis.** We list the genes corresponding to the CpG dinucleotides that are in the top 20 according to either the change in the esitmated age, $|\Delta a|$, the $m$-reach, $r_m$, or the average control centrality, $\langle c \rangle$. The background colouring of the cells indicates the relative magnitude of the given value compared to the other CpG dinucleotides.

| gene | $|\Delta a|$ | $r_m$ | $\langle c \rangle$ | gene | $|\Delta a|$ | $r_m$ | $\langle c \rangle$ |
|---|---|---|---|---|---|---|---|
| SCGN | 5.74 | 0.48 | 0.71 | PQLC1 | 1.56 | 0.84 | 0.8 |
| BAZ2A | 5.54 | 0.86 | 0.78 | C16orf65-cg0 | 1.51 | 0.84 | 0.8 |
| UCKL1 | 5.07 | 0.9 | 0.81 | C3orf75-cg1 | 1.5 | 0.83 | 0.78 |
| AGBL5 | 5.02 | 0.87 | 0.79 | PAPOLG | 1.5 | 0.94 | 0.8 |
| CEBPD | 4.09 | 0.62 | 0.79 | KDM3A | 1.3 | 0.71 | 0.8 |
| RXRA | 3.93 | 0.74 | 0.79 | CXADR | 1.29 | 0.67 | 0.8 |
| C19orf30-cg0 | 3.93 | 0.54 | 0.71 | GPR68 | 1.25 | 0.81 | 0.8 |
| NHLRC1 | 3.78 | 0.29 | 0.67 | DNASE2 | 0.98 | 0.88 | 0.8 |
| PAWR-cg0 | 3.2 | 0.58 | 0.79 | ABCA3 | 0.9 | 0.73 | 0.79 |
| VGF | 2.97 | 0.39 | 0.71 | MN1 | 0.88 | 0.86 | 0.8 |
| DPP8 | 2.87 | 0.5 | 0.71 | MIB1 | 0.84 | 0.86 | 0.79 |
| C21orf63 | 2.72 | 0.66 | 0.72 | ELAC2 | 0.76 | 0.87 | 0.78 |
| TNFRSF13C | 2.45 | 0.66 | 0.79 | C7orf44 | 0.66 | 0.84 | 0.79 |
| TSSK6 | 2.43 | 0.71 | 0.78 | LEPRE1 | 0.39 | 0.81 | 0.8 |
| NR2F2 | 2.4 | 0.49 | 0.66 | PPP1R16B | 0.39 | 0.86 | 0.74 |
| ABHD14A-cg0 | 2.32 | 0.37 | 0.71 | ZHX1 | 0.31 | 0.81 | 0.79 |
| CSNK1D | 2.31 | 0.42 | 0.64 | CYFIP1 | 0.24 | 0.84 | 0.81 |
| AFF1 | 2.31 | 0.57 | 0.72 | C7orf55 | 0.24 | 0.81 | 0.8 |
| EIF3M | 2.3 | 0.82 | 0.74 | WFS1 | 0.21 | 0.76 | 0.8 |
| C19orf30-cg1 | 2.07 | 0.39 | 0.71 | BBS5 | 0.2 | 0.85 | 0.76 |
| C14orf176 | 1.64 | 0.91 | 0.8 | CATSPERG | 0.16 | 0.95 | 0.8 |

*genes* we shortly describe the role and function of a couple of the notable genes from the table, collected from the recent scientific literature.

The methylation sites and the related genes involved Horvath's clock bear remarkable power for age estimation but that not necessarily means that they control the aging process. Our network analysis identified a hierarchical control structure, but keep in mind that this study was limited to the Horvath's clock 353 CpG dinucleotides, so the role of the top genes' may be less important in an absolute sense and more influential control genes may remain hidden. The system we examined can be viewed as a sub-graph of a much larger network, i.e., new version of methylation microarrays can measure metylation at 850,000 CpG dinucleotides and whole genome bisulfite sequencing may observe millions. Thus, it is quite plausible that if more and more CpG dinucleotides are involved in a similar network based study, the additional interactions with the rest of this larger system might change the roles of the nodes in the focus of the present paper.

Due to the huge computational requirements, the repetition of our analysis on e.g., the network between the 850,000 CpG dinucleotides of current microarrays was not feasible, however, extension of the work on larger CpG dinucleotid networks is subject of further research. In the mean time, we have also examined the methylation network corresponding to two further epignetic clocks, the Skin-Blood clock [33] (with roughly the same number of CpGs as Horvath's clock) and Hannum's clock (smaller than Horvath's clock), as well as methylation networks consisting of randomly chosen CpGs with the same size as Horvath's clock, and the methylation network for the CpGs in Horvath's clock based on the data set studied by Lehne et al. in Ref [62]. According to the results detailed in the S1 Text, these networks display hierarchical and control properties quite similar to what we have observed for Horvath's clock based on the data studied by Hannum et al. in Ref. [28]. In addition, the alternative hierarchy obtained from the data set studied by Lehne et al. showed a reasonably high similarity with the hierarchy described in Figs 2 and 3. These findings indicate that methylation networks obtained in our framework can be expected to be hierarchical (with coupled control properties) in general. Furthermore, our analysis of networks where the top (or bottom) 10% of the CpGs from the hierarchy of Horvath's clock were mixed with randomly chosen CpGs revealed that the position of the nodes in the hierarchy is showing a considerable conservation when the constituents of the network are varied. Based on this it is quite plausible that the nodes found to be close to the top in small hierarchies may play a more important role than the average in larger methylation networks as well.

In conclusion, we analysed the methylation network between the CpG dinucleotides appearing in Horvath's clock. The inference of the connections between the CpGs building up this network is based on that many ageing effects (such as the age-related myeloid skew [40], T cell exhaustion [41]), polycomb target hypermethylation [20], bivalent domain hypermethylation [42], etc) result in coordinated changes in the methylation profile. The links obtained in our approach can be interpreted as a simple linear fit of the correlated differences between the methylation profiles in the data. According to our analysis based on the $m$-reach and the GRC, the studied network is substantially more hierarchical compared to a random graph with the same degree distribution. In addition, the network displays interesting control properties as well, e.g., the nodes at the top of the hierarchy tend to have larger control centrality as well. We also studied the effect of methylation level perturbations on the estimated age in a framework where the perturbations were propagated over the methylation network. According to our analysis, the resulting modifications to the overall methylation pattern seemed to be more aligned with the natural direction of ageing in a high dimensional representation of the methylation state compared to isolated methylation changes. Furthermore, the network framework scenario can also provide a significantly larger change in the predicted age. Finally, the nodes

with large potential for achieving a notable change in the predicted age are more frequent in the top part of the hierarchy according to $m$-reach, and seem to have a large control centrality as well. Thus, when perturbing the methylation network at nodes having an important role from the topological point of view, the resulting effect in the predicted age is likely to be more intense compared to the response obtained by modifying the methylation of ordinary nodes. These findings indicate that the network approach can bring new insight into methylation-related studies, providing a very interesting direction for further research.

## Methods

### Methylation data

Our study is based on the publicly available methylation data published by G. Hannum et. al in Ref. [28]. The authors used the data to build a DNA methylation-based age estimator, so they performed genome-wide methylomic profiling of a large number of individuals spanning a wide age range. The data set consists of samples from a healthy population of 426 Hispanic and 230 Caucasian individuals, aged 19 to 101 with a median age of 65 years. The study included 338 female and 318 male participants. The samples were taken as whole blood and processed according to the standard protocol of the Illumina Infinium HumanMethylation450 BeadChip, which quantifies DNA methylation levels as a fraction between zero and one (also known as beta value). The BeadChip measures the methylation levels in over 480k CpG sites, including the 353 CpG dinucleotides of Horvath's clock. In our study we only used this subset of the methylation data as a $353 \times 656$ table, and additionally we extracted the age of the individuals from the metadata as a list of 656 integers. The complete methylation profiles and the metadata can be found in NCBI's Gene Expression Omnibus (GEO) under accession number GSE40279.

### Lasso-CV regression

When carrying out Lasso regression in general, the objective is to solve

$$\min_{\beta}\left\{\frac{1}{N}\|\boldsymbol{y} - \boldsymbol{X}\boldsymbol{\beta}\|_2 + \alpha\|\boldsymbol{\beta}\|_1\right\}, \tag{6}$$

where $\boldsymbol{y}$ denotes the outcome (response variable), $\boldsymbol{X}$ is the matrix of the regressors (feature variables), $\boldsymbol{\beta}$ corresponds to the regression coefficients, $\alpha$ is a parameter, and $\|\cdot\|_1$ and $\|\cdot\|_2$ denote the $L^1$ and $L^2$ norms, respectively. The advantage of this approach is that due to the second term, a large fraction of the regression coefficients become exactly zero, leaving only the really relevant predictor variables in the game.

In order to obtain more reliable results, the general technique of cross-validation can be combined with the idea of Lasso, hence the approach is usually referred to as Lasso cross-validation regression. The basic idea is to distribute the data into a number of 'folds' at random, and carry out the above regression on each fold separately. The results are then tested on the whole data set, and the best fit is chosen as the final solution to the regression problem. In our studies we used a 10 fold cross-validation when applying Lasso regression to infer connections between the CpG dinucleotides of Horvath's clock, corresponding to a standard choice for the number of folds.

### Finding the optimal weight threshold based on network efficiency

As mentioned in Results, the methylation network obtained from Lasso-CV is still relatively dense (although a considerable part of the regression coefficients is zero), with the average

degree taking a value of $\langle k \rangle = 97.6$. In order to make the network sparser, a standard procedure in complex network theory was used to apply a link weight threshold $w^*$, throwing away the weak connections where $w_{ij} < w^*$, and to keep only the strongest, most relevant links. For choosing the optimal value for $w^*$, we use the general method introduced in Ref. [60], designed to find the best link weight threshold in dense biological networks.

The key idea of this method is to monitor the changes in a quality function based on network efficiency as a function of the link density in the network, and choose the settings where this function is maximal. The quality function is given by

$$J = \frac{E_g + E_l}{\rho_L}, \tag{7}$$

where $E_g$ and $E_l$ denote the global- and the local efficiency of the network, and $\rho_L$ is equal to the link density ($\rho_L = L/[N(N-1)]$, with $L$ denoting the total number of links). The global efficiency is defined as the average of the inverse distance between all node pairs in the network [63], whereas the local efficiency is the average of $E_g$

$$E_g = \frac{1}{N(N-1)} \sum_{i \neq j} \frac{1}{d_{ij}}, \tag{8}$$

where $d_{ij}$ stands for the length of the shortest path from $i$ to $j$, and based on that, can take high values when the majority of the nodes are close to each other in the network sense. The local efficiency is given by

$$E_l = \frac{1}{N} \sum_{i=1}^{N} E_g(i), \tag{9}$$

where $E_g(i)$ denotes the global efficiency calculated for the sub-graph between the neighbours of node $i$ (where node $i$ itself is absent from the sub-graph).

In Fig 9 we show $J$ as a function of $w^*$, displaying a single maximum at $w^* = 0.641$. According to that, we used this value for thresholding the connections in the methylation network.

## Hierarchy and reach

To analyse the hierarchical properties of the obtained methylation network we can use the concept of the Global Reaching Centrality hierarchy measure [52, 55, 61], which is based on the reach of the nodes. The basic idea behind this approach is that for leaders at the top of a hierarchy it should be easy to reach the rest of the nodes via relatively short paths, whereas this is not the case for the bottom nodes. Thus, by comparing the fraction of nodes reachable in at most $m$ steps from a given node (called as the $m$-reach of the node) we can more or less judge which nodes should be positioned high in the hierarchy (the ones with a large $m$-reach), and which nodes are supposed to be at the lower levels (the nodes with small $m$-reach values). In addition, based on the inhomogeneity of the $m$-reach distribution we can also quantify the extent of hierarchy in the organization of the network. The Global Reaching Centrality is defined along this idea as

$$\mathrm{GRC}(m) = \frac{1}{N-1} \sum_{i=1}^{N} \left[ r_{m,\max} - r_m(i) \right], \tag{10}$$

where $r_m(i)$ is the $m$-reach of node $i$, and $r_{m,\max}$ is the maximum value of the $m$-reach in the network. The original version of the GRC in Ref. [61] was defined using the total reach corresponding to $m = \infty$, however later it turned out that lower values of $m$ might work better in a

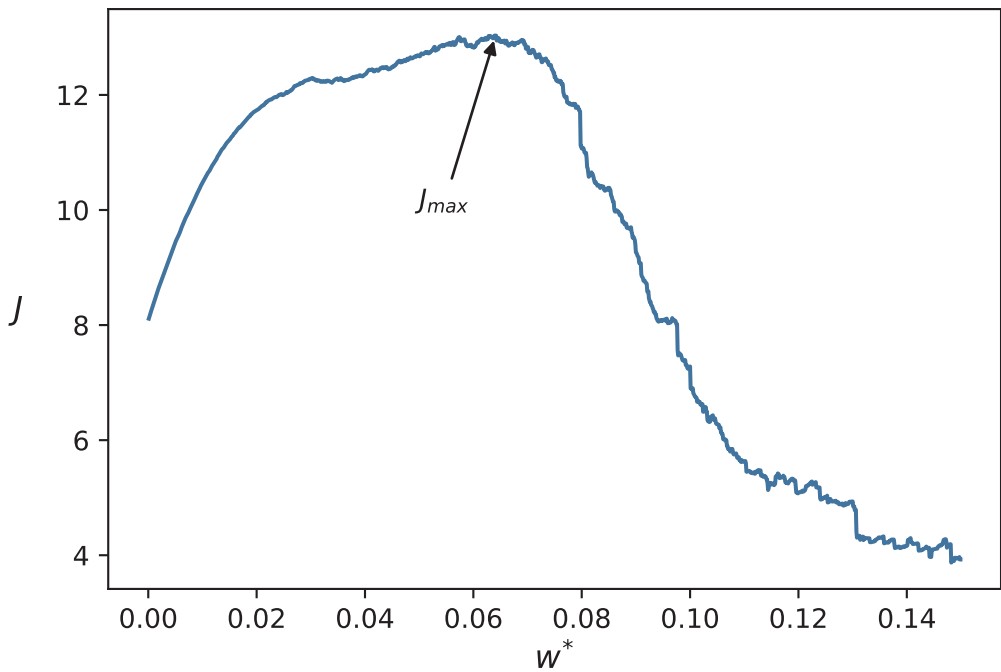

**Fig 9. Efficiency as a function of the weight threshold.** The efficiency $J$ obtained from (7) as a function of the weight threshold $w^*$. According to the plot, the optimal value of $w^*$ is at $w^* = 0.641$.

number of networks [52, 55]. A large value of the $\mathrm{GRC}(m)$ is obtained for inhomogenous $m$-reach distributions, indicating a strong hierarchical organization in the network structure, whereas low values of the $\mathrm{GRC}(m)$ usually corresponds to non-hierarchical networks where the $m$-reach of the nodes is more or less the same throughout the system.

When comparing the $\mathrm{GRC}(m)$ measured in the original methylation network to random networks with the same degree distribution, the basic idea is to take the configuration model [64] as a sort of 'base line', and examine whether we see deviations from this random graph base line in the real network. In the configuration model the a random graph ensemble is defined based on the degree sequence (corresponding to the list of degrees appearing in the network), and all possible realisations of graphs with the given degree sequence are considered equally probable. In practice, samples from the ensemble are generated by randomising the original network, where the degrees of the nodes are left intact. In our studies we used link randomisation, where in each step a pair of links are chosen at random, and one end on these links is swapped. The total number of randomisation steps was set such that the average number of rewiring per link was 10 for each random graph sample.

## Network control and control centrality

In the linear control theory of networks we assume a time dependent state variable $x_i(t)$ assigned to each node $i$, that are governed by the differential equation

$$\frac{\mathrm{d}x_i}{\mathrm{d}t} = \sum_{j=1}^{N} A_{ij} x_j(t) + \sum_{q=1}^{Q} B_{iq} u_q(t),\qquad(11)$$

where $A_{ij}$ is corresponding to the adjacency matrix of the network, and $B_{iq}$ is an $N$ by $Q$ external input matrix, where we the input variables $u_q(t)$ can be chosen at will. The nodes actually

receiving external input (for which at least one $B_{iq}$ is non-zero) are called as driver nodes. The system is controllable, if by appropriate choice of the input variables $u_q$ we can drive the system from any initial state to any desired state [65–67]. One of the fundamental results of structural controllability is that a directed weighted network is controllable if and only if its unweighted counterpart is controllable [58] (except for 'singular' distribution of the link weights, which however can be considered as a zero measure set among all possible link weight assignments). Thus, when studying the control properties of a directed real network, we can assume all the non-zero weights to be equal to 1, thereby effectively turning the network into an un-weighted one, and concentrate solely on the network structure.

Structural controllability is very closely related to the matching problem in networks [58]. Matching in its most intuitive and original form is defined for bipartite graphs, where we have two node sets (e.g., 'top' and 'bottom' or 'left' and 'right'), and links can only connect nodes from different sets. A matching is a subset of non-adjacent links, where link adjacency means a common end point. This non-adjacency property means that these links provide a unique one-to-one correspondence between the involved nodes (hence the name matching). In a perfect matching we can find a match for every node (where obviously, the two sets of nodes have to be of equal size to make this possible). In general we can look for the maximal matching, where the number of links in the matching is maximal. This is usually not unique, i.e., multiple different non-adjacent link sets may have the same maximal number of links. An efficient way to find a possible maximal matching of a bipartite graph is given by the Hopcroft-Karp algorithm [68].

The concept of matching can be extended from bipartite graphs to a general directed network by taking the nodes with at least one out-link and treating them as the 'top' set, whereas the nodes with at least in-link are considered as the 'bottom' set. Although the nodes with both in- and out-links appear in both sets, this is not a problem, since there is a one to one correspondence between the links in the original directed network and the links in the newly defined bipartite graph. Once we have found a maximal matching in the bipartite graph, we can map it back onto the original directed network, where the nodes with an incoming matching link are considered as matched nodes and nodes without any incoming matching link are considered un-matched.

According to Ref. [58], for any directed network with a perfect matching the minimum number of driver nodes $n_D$ needed to fully control the network is $n_D = 1$ (with an arbitrary choice of the driver node), whereas if the network has no perfect matching, $n_D$ is equal to the number of unmatched nodes for any maximal matching, and the driver nodes are actually corresponding to the unmatched nodes. Therefore, the controllability of a network and the minimum number of driver nodes needed to fully control the system can be obtained by solving the matching problem on the same graph. However, since the maximal matching is usually not unique, it can easily happen that the same node is considered as a driver node for one particular maximal matching, and as a controlled node for another maximal matching.

In order to quantify the importance of a given node from the point of view of control, the concept of control centrality was introduced in Ref. [59], circumventing the above mentioned ambiguity in the actual role of the node over the different possible maximal matchings. To calculate the control centrality $C(i)$ for a given node $i$, first we have to consider the sub-graph that is reachable from $i$ (by following recursively the out-links), and add an incoming external control link to $i$, making it a driver node. The control centrality of $i$ is then given by the maximum number of controlled nodes in the reachable sub-graph (the number of links in the maximum matching for this sub-graph). The intuitive meaning of $C(i)$ is that it corresponds to the maximum number of nodes that we can drive in the system by controlling the given node using an external signal. The relative control centrality used in

the present work is simply $C(i)$ normalised by the number of nodes as $c(i) = C(i)/N$. In our studies of the control centrality we used the Hopcroft-Karp algorithm [68] for calculating the number of links in the maximum matching for the reachable sub-graphs of the CpG dinucleotides in the methylation network.

## Biological role of top identified genes

One CpG dinucleotide (cg02047577), in the *UCKL1* gene, was common in all three top 20 lists. The protein encoded by the *UCKL1* gene is a uridine kinase and it is involved in the pyrimidine metabolism pathway. Methylation at this CpG site in the *UCKL1* gene negatively correlates with age and it has been shown that the gene expression of *UCKL1* is increasing during ageing in ovary [69] and skin [70].

Worth highlighting the *BAZ2A* gene which is an essential component of the NoRC complex. This complex mediates the silencing of a fraction of ribosomal DNA [71] and heterochromatin formation at centromeres and telomeres [72]. In the complex, BAZ2A plays a central role by being involved in the recruitment of chromatin modifying enzymes, such as HDAC1, DNMTs and ISWI-ATPase nucleosomal constriction machinery, resulting in collaborative silencing [73].

A comprehensive methylation analysis of CpG sites in DNA from blood cells identified 102 age-associated CpGs and showed a positive correlation between hypermethylation of *SCGN* and age [74]. The *SCGN* gene is involved in the Ca, cAMP and Lipid Signaling pathway which regulates various cellular functions, including cell growth, cell differentiation, gene transcription and protein expression [75].

In a study examining DNA methylation of human brain tissue samples, a CpG site (cg14424579) in the *AGBL5* gene showed a highly significant positive correlation with chronological age [76]. The protein, encoded by the *AGBL5* gene, has a deglutamylase activity and it has been indicated to be involved in the regulation of the immune response to DNA viruses [77] which may be related to the well known age-associated changes of the immune system and the increased susceptibility of elderly individuals to infectious diseases. TNFRSF13C (Tumour Necrosis Factor Receptor Superfamily Member 13C) is the principal receptor required for BAFF-mediated B-cell maturation and survival which may be connected to the age-associated changes in the B-cell lineage [78]. The protein encoded by the *TNFRSF13C* gene plays a role in multiple essential pathways, such as the Cytokine-cytokine receptor interaction pathway and the NF-$\kappa$B pathway. The latter is known to be one of the key mediators of ageing activated by genotoxic, oxidative and inflammatory stress [79].

The protein encoded by the *CEBPD* gene is an important transcription factor regulating the expression of genes involved in immune and inflammatory responses and may be involved in the regulation of genes associated with activation and/or differentiation of macrophages. Furthermore, the RXRA (Retinoid X Receptor Alpha) protein acts as a transcription factor involved in the regulation of gene expression in various biological processes, for example, plays a role in the attenuation of the innate immune system in response to viral infections [80] and involved in the regulation of calcium signalling and cellular senescence [81] which is a known hallmark of ageing. The protein encoded by the *NR2F2* gene is a ligand-activated transcription factor that is involved in the regulation of many different genes within various pathways, such as the Oct4 in Mammalian ESC Pluripotency and the Regulation of Telomerase. Interestingly, CEBPD, RXRA and NR2F2 are super-enhancer-associated transcription factors identified in multiple cell types [82]. Super-enhancers are clusters of enhancers in the mammalian genome bound by a number of transcription factors and coactivators driving high-level expression of key regulators of cell identity, although showing exceeding vulnerability to

perturbation of their components [83]. In a recent study, it has been shown that site-specific demethylation of CEBPB/D-dependent adipogenic super-enhancers mediated by the GADD45$\alpha$-ING1 complex directly controls energy metabolism and ageing in mice [84]. A subset of super-enhancer-associated transcription factors, called the Yamanaka factors (Oct3/ 4, Sox2, Klf4, c-Myc), have a critical role in the regulation of the developmental signalling network [85] and the overexpression of these factors in mouse fibroblasts induces them to become pluripotent stem cells that can differentiate into almost any other cell type [86]. Reprogramming of cells with the Yamanaka factors causes epigenetic changes and the molecular markers of ageing can be slowed down and even reversed by reprogramming [87]. In addition, the Horvath epigenetic clock is reset to zero in induced pluripotent stem cells, such as in embryonic stem cells [5]. Taking this together we may speculate that DNA methylation changes in CEBPD, RXRA and NR2F2 as super-enhancer-associated transcription factors may contribute to the dysregulation of developmental genes and the loss of cellular identity observed during ageing.

## Supporting information

**S1 Text. Figure A**: "Expected changes in the estimated age" **Figure B**: "Hierarchy of the methylation network based on the Skin-Blood clock" **Figure C**: "Hierarchy of the methylation network based on the Hannum's clock" **Figure D**: "Control centrality and reach in the network defined based on the Skin-Blood clock" **Figure E**: "Control centrality and reach in the network defined based on the Hannum's clock" **Figure F**: Scatter plot of expected change in the estimated age as a function of the the $m$-reach and the control centrality for the network based on the Skin-Blood clock. **Figure G**: Scatter plot of expected change in the estimated age as a function of the the $m$-reach and the control centrality for the network based on the Hannum's clock. **Figure H**: Hierarchy of the methylation networks based on randomly chosen CpGs. **Figure I**: Control centrality and reach in methylation networks based on randomly chosen CpGs. **Figure J**: Hierarchy of the methylation networks based on the data set studied by Lehne et. al. **Figure K**: Control centrality and reach in methylation networks based on the data set studied by Lehne et. al. **Figure L**: Scatter plot of the expected change in the estimated age as a function of the the $m$-reach and the control centrality for the network based on the data studied by Lehne et. al. **Figure M**: Correlation between the $m$-reach obtained in the networks based on the Hannum et. al and on the Lehne et al. dataset. **Figure N**: Distribution of the top 10% of the CpGs according to the Lehne et al. hierarchy in the hierarchy based on the Hannum et al. network. **Figure O**: Distribution of the top 10% of the CpGs according to the Hannum et al. hierarchy in the hierarchy based on the Lehne et al. network. **Figure P**: Age distribution of the patients in the studied data sets.
(PDF)

## Author Contributions

**Conceptualization:** Gergely Palla, Péter Pollner, Béla Molnár, István Csabai.

**Data curation:** András Major.

**Formal analysis:** Gergely Palla, Péter Pollner, Judit Börcsök, István Csabai.

**Investigation:** Gergely Palla, Péter Pollner, Judit Börcsök, István Csabai.

**Methodology:** Gergely Palla, Péter Pollner, István Csabai.

**Supervision:** István Csabai.

**Validation:** Gergely Palla.

**Visualization:** Gergely Palla.

**Writing – original draft:** Gergely Palla.

**Writing – review & editing:** Gergely Palla, Péter Pollner, Judit Börcsök, András Major, István Csabai.

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
