## [Decision Letter · Decision Letter 0]

18 Jun 2020

Dear Dr Pollner,

Thank you very much for submitting your manuscript "Hierarchy and control of ageing-related methylation networks" for consideration at PLOS Computational Biology.

As with all papers reviewed by the journal, your manuscript was reviewed by members of the editorial board and by several independent reviewers. In light of the reviews (below this email), we would like to invite the resubmission of a significantly-revised version that takes into account the reviewers' comments.

We cannot make any decision about publication until we have seen the revised manuscript and your response to the reviewers' comments. Your revised manuscript is also likely to be sent to reviewers for further evaluation.

Sincerely,

Ilya Ioshikhes

Associate Editor

PLOS Computational Biology

Douglas Lauffenburger

Deputy Editor

PLOS Computational Biology

Reviewer's Responses to Questions

**Comments to the Authors:**

Reviewer #1: Please see attachment

Palla et al. have produced a manuscript entitled “Hierarchy and control of ageing-related methylation networks.” In this paper they have extracted an interaction network from the CpGs employed in the Horvath clock. Unsurprisingly, this shows some hierarchical organisation is present. They then go on to discuss how modifying the clock will led to age ‘reversal’. Unfortunately, there are significant issues in the design and conclusions drawn from this study, due to imprecise understanding of the ageing biology of the epigenome, as well as the construction and interpretation of the Horvath clock. The researchers have performed a network analysis focused only the 353 CpGs from this specific clock, without acknowledging that these CpGs themselves are not uniquely special in regard to their functionality. The discussion of age-reversal gives these methylation sites a definitely active role in the ageing process which they do not possess. My concerns are listed below.

Major

1. The statements in the Abstract that it is “plausible to assume that by proper adjustment of these switches age may be tuned” and that “biological clock can be changed or even reversed” – are counter to the current understanding of the field and imply that the clock itself is driving ageing rather than a ‘biomarker’ of the ageing process and the plethora of ageing-related changes it is capturing1. The clock itself is used to measure the impact of potential interventions2.

2. Furthermore, the statement that “adjustment of one leads to a cascade of changes at other sites” is not surprising if one understands what biological and connected epigenetic changes will be represented, as in this case of blood tissue derived DNA3.

3. The statement in Abstract and elsewhere that ‘we locate the most important CpGs’ ignores the fact that they limit their analysis to only the 353 CpG from the total DNA methylome of 28 million CpGs to begin with. As Horvath has stated there is no evidence that the CpGs in the Horvath clock are especially functional over and above many other CpGs and reasonable clocks can be constructed from even a random selection of CpGs - there are abundant potential CpGs that can be exploited in clocks3. The statement “largest influence” and “which may also play a crucial role in the process of ageing” (Introduction, Line 94) again implies these small fraction of 353 CpGs are uniquely special4.

4. Age-related change in DNA methylome is in fact widespread with up to 15 – 30% of all CpG sites in the genome associated with age-related changes and these are not all called ‘clock CpGs’ (Introduction line 18). Change can be random fashion due to epigenomic drift5, directional, or show increased variability with age6. Also, the statement regarding the directionality of “clock CpGs that are hypermethylated” (Introduction line 35) is an oversimplification. Teschendorff et al. identified an enrichment in an early promoter-focused array for age-related CpGs that were hypermethylating in the Targets of Polycomb Target gene promoters, but genome-wide hypomethylation predominates. Both hypo- and hypermethylated loci contribute to the various published clocks.

5. The statement in the Introduction that there are “connections between the CpGs themselves’ (line 75) is as expected. Clearly all well-known ageing effects lead to co-ordinated changes across the entire DNA methylome – these include those driven by cell-type specific epigenomics where changes in cell proportion will led to variation (including the age-related myeloid skew7, T cell exhaustion)8, polycomb target hypermethylation9, bivalent domain hypermethylation10, etc. These known systemic effects will be seen as networks of age-related change.

6. Distinct biological processes drive the observed age-related hypermethylation and hypomethylation. Furthermore, the baseline DNA methylation state is strongly driven by genetics being highly CpG density dependent11.

7. The statement (line 53) that “we cannot really point out any of these CpGs as being more important than others” is as completely expected in the way that the elastic net regression Horvath clock was designed. CpGs were selected not for their individual strength but chosen for their power to work collectively to parsimoniously capture ageing over the lifecourse. In fact, this is clearly demonstrated by the fact the strongest and most robust individual CpG pan-tissue changes from the ELOVL2 locus12,13 were not included in the clock. Additionally, an accurate clock has been devised using just 3 CpGs14.

8. The discussion of “control properties” of CpGs is consistent with the Elastic Net picking those CpGs that work well together. Thus, the results regarding network identification and properties have ignored this and the limited CpGs this has been exacted from e.g. Results (line 112). Why were not all the ~850,000 CpGs from the EPIC array analysed in the network analysis rather than just 353? Conclusion statements regarding how a “network approach can bring new insight into methylation-related studies, providing a very interesting direction for further research” (Line 389) are clearly limited when restricted to only these 353 CpGs and known biology not taken into account.

9. The authors need to explain and understand more precisely what the concept of ‘biological age’ and predicators of this represent15. The initial Horvath clock was devised as an attempt at a ‘pan-tissue’ clock (which it was highly successful in although caveats remain16,17). It is in fact a ‘composite’ clock3 capturing both forensic and biological age but neither perfectly. The authors need to understand and integrate the current knowledge and issues regarding DNA methylation clocks - as discussed recently by the epigenomics community4.

10. The statements regarding “Modifying the predicted age by perturbing the methylation network’ need to be put in the context that they are interpreting a ‘biomarker’ of biological ageing.

11. Unclear what “more aligned with the 'natural direction of ageing'.” (Line 283) means biologically?

12. In the Discussion the statement ‘Horvath's clock is showing non-trivial hierarchical and control properties’ – how is this unexpected? Furthermore, how would that be different from a random selection of array-derived CpG probes?

13. The statements regarding the functional implications of individual CpGs in the Discussion need to be more clearly caveated8.

14. In the Conclusion (line 374) the statement “substantially more hierarchical compared to a random Graph” does not take into consideration the biological nature of these data.

Minor

1. English needs correcting throughout manuscript

2. Abstract – Grammar - “…biomarkers of ageing”

3. “specific CpG pairs” line 20 – CpG ‘dinucleotides’ is usually stated as more precise

4. Spelling line 33 - DNA methylation

5. Gene names are by convention written in italics – e.g. UCKL1 gene (line 314) etc.

1. Horvath, S. & Raj, K. DNA methylation-based biomarkers and the epigenetic clock theory of ageing. Nat Rev Genet 19, 371-384 (2018).

2. Fahy, G.M. et al. Reversal of epigenetic aging and immunosenescent trends in humans. Aging Cell 18, e13028 (2019).

3. Field, A.E. et al. DNA Methylation Clocks in Aging: Categories, Causes, and Consequences. Mol Cell 71, 882-895 (2018).

4. Bell, C.G. et al. DNA methylation Aging Clocks: Challenges & Recommendations. Genome Biology (2019).

5. Feil, R. & Fraga, M.F. Epigenetics and the environment: emerging patterns and implications. Nature reviews. Genetics 13, 97-109 (2011).

6. Slieker, R.C. et al. Age-related accrual of methylomic variability is linked to fundamental ageing mechanisms. Genome Biol 17, 191 (2016).

7. Rimmelé, P. et al. Aging-like Phenotype and Defective Lineage Specification in SIRT1-Deleted Hematopoietic Stem and Progenitor Cells. Stem Cell Reports 3, 44-59 (2014).

8. Lappalainen, T. & Greally, J.M. Associating cellular epigenetic models with human phenotypes. Nat Rev Genet 18, 441-451 (2017).

9. Teschendorff, A.E. et al. Age-dependent DNA methylation of genes that are suppressed in stem cells is a hallmark of cancer. Genome Res 20, 440-6 (2010).

10. Rakyan, V.K. et al. Human aging-associated DNA hypermethylation occurs preferentially at bivalent chromatin domains. Genome Res 20, 434-9 (2010).

11. Baubec, T. & Schübeler, D. Genomic patterns and context specific interpretation of DNA methylation. Current Opinion in Genetics & Development 25, 85-92 (2014).

12. Garagnani, P. et al. Methylation of ELOVL2 gene as a new epigenetic marker of age. Aging Cell 11, 1132-4 (2012).

13. Slieker, R.C., Relton, C.L., Gaunt, T.R., Slagboom, P.E. & Heijmans, B.T. Age-related DNA methylation changes are tissue-specific with ELOVL2 promoter methylation as exception. Epigenetics & Chromatin 11, 25 (2018).

14. Weidner, C.I. et al. Aging of blood can be tracked by DNA methylation changes at just three CpG sites. Genome Biol 15, R24 (2014).

15. Jylhävä, J., Pedersen, N.L. & Hägg, S. Biological Age Predictors. EBioMedicine 21, 29-36 (2017).

16. Horvath, S. et al. Epigenetic clock for skin and blood cells applied to Hutchinson Gilford Progeria Syndrome and ex vivo studies. Aging (Albany NY) 10, 1758-1775 (2018).

17. Zhang, Q. et al. Improved precision of epigenetic clock estimates across tissues and its implication for biological ageing. Genome Med 11, 54 (2019).

Reviewer #2: The study by Palla et al., entitled “Hierarchy and control age-related methylation networks”, revealed that the age-related CpGs are interconnected, with dynamic methylation change on one CpG probably leading to a cascade of changes at the other sites. It provided a framework to explore the key methylation sites during ageing process, which might be applied to other biomarkers/biological processes. This study is interesting but remains too preliminary, as the authors only focused on the 353 Horvath’s “clock CpGs”. To better understand the issue raised in the study, a comprehensive analysis of CpGs involved in ageing and age-related phenotypes/diseases should be considered by collecting more methylation data. In this case, the manuscript needs to be revised thoroughly before considering for publication.

Major concerns:

1) The Introduction section is poorly summarized. Authors need simplify the content and clarify the background and purpose of the study.

2) Evidence supporting the leading roles of identified CpGs during ageing is insufficient. For example, the training model should be tested in multiple datasets. And, if possible, it will be great if some functional assays are performed.

3) How shall we view the key CpGs’ roles in ageing? It’s hard to determine whether the methylation status is the result or the cause of ageing process. Authors should discuss this in Discussion section.

4) Whether the training model can be used to scan the key CpGs that control various biological processes?

5) Is there any correlation between a certain CpG’s methylation status and its hierarchy level? (For example, sites located on higher levels may also have lower methylation values.)

6) The network is based on only 353 sites. When considering the CpGs across whole genome, the perturbation results may be different or even opposite. Authors may consider adding some data or results to demonstrate the robustness of the perturbation results. Generally, authors should, at least, provide evidence showing that the 353 “clock sites” are less affected by “non-clock sites”.

Minor concerns:

7) The Formula 4 doesn’t render properly in the ms for reviewers.

8) The Discussion section seems too long, it talks too much on genes’ functions. Authors may move and summarize these contents into the Results section.

**Have all data underlying the figures and results presented in the manuscript been provided?**

Reviewer #1: Yes

Reviewer #2: Yes

PLOS authors have the option to publish the peer review history of their article (what does this mean?). If published, this will include your full peer review and any attached files.

Reviewer #1: No

Reviewer #2: No
---

## [Decision Letter · Decision Letter 1]

21 Sep 2020

Dear Dr. Pollner,

Thank you very much for submitting your manuscript "Hierarchy and control of ageing-related methylation networks" for consideration at PLOS Computational Biology.

As with all papers reviewed by the journal, your manuscript was reviewed by members of the editorial board and by several independent reviewers. In light of the reviews (below this email), we would like to invite the resubmission of a significantly-revised version that takes into account the reviewers' comments.

We cannot make any decision about publication until we have seen the revised manuscript and your response to the reviewers' comments. Your revised manuscript is also likely to be sent to reviewers for further evaluation.

It is mandatory that the reviewers will be satisfied by the revisions made in this round. 

Sincerely,

Ilya Ioshikhes

Associate Editor

PLOS Computational Biology

Douglas Lauffenburger

Deputy Editor

PLOS Computational Biology

Reviewer's Responses to Questions

**Comments to the Authors:**

Reviewer #1: See Attachment

Response to Reviewer 1:

We thank the Referee for the careful and detailed examination of the manuscript and the extremely valuable comments, which indeed have helped making our paper better. We are truly grateful for the 17 bibliographic references included in the report that we now also cited in the revised version of the paper. Our detailed answers to the points raised are the following

Palla et al. have produced a manuscript entitled “Hierarchy and control of ageing-related methylation networks.” In this paper they have extracted an interaction network from the CpGs employed in the Horvath clock. Unsurprisingly, this shows some hierarchical organisation is present. They then go on to discuss how modifying the clock will led to age `reversal'. Unfortunately, there are significant issues in the design and conclusions drawn from this study, due to imprecise understanding of the ageing biology of the epigenome, as well as the

construction and interpretation of the Horvath clock. The researchers have performed a network analysis focused only the 353 CpGs from this specific clock, without acknowledging that these CpGs themselves are not uniquely special in regard to their functionality. The discussion of age-reversal gives these methylation sites a definitely active role in the ageing process which they do not possess.

My concerns are listed below.

Major

1. The statements in the Abstract that it is “plausible to assume that by proper adjustment of these switches age may be tuned" and that “biological clock can be changed or even reversed" are counter to the current understanding of the field and imply that the clock itself is driving ageing rather than a `biomarker' of the ageing process and the plethora of ageing-related changes it is capturing [1]. The clock itself is used to measure the impact of potential interventions [2].

We thank for the referee to point out this possible misunderstanding. It is widely accepted and demonstrated by various epigenome editing studies that DNA methylation is one of the most important factors that control gene expression, activation and splicing, hence many of the biological processes of the living systems. We agree that methylation is only one of the possible factors that control the ageing process and also that the 353 CpG-s in Horvath's clock give only a very small subset of them. We have reworded the abstract and explicitly stated that correlation is not equivalent to causation. We have also acknowledged that we demonstrate our approach only on a small set of CpG sites and to get biologically relevant control nodes, the analysis should be extended to all methylation sites. In the revised version of the manuscript we mention the use of epigenetic clocks for the measurement of the impact of a thymus regeneration protocol as described in Ref.[2], whereas Ref.[1] was cited already in the original submission.

• Good to acknowledge that causation differs from correlation. The point regarding ageing also specifically refers to the biological processes that are observed in blood as DNA methylation variation – as listed in regard to point 5 below.

2. Furthermore, the statement that “adjustment of one leads to a cascade of changes at other sites" is not surprising if one understands what biological and connected epigenetic changes will be represented, as in this case of blood tissue derived DNA [3].

We agree, living things are complex interconnected systems. One of our goals with this paper was to emphasise this fact and to make the first step from the widely used linear models toward network model that may capture some of the complexities. We have reworded the cited sentence to avoid the false interpretation, and inserted a citation to Ref[3] from the referee report into the Introduction.

• The point is not the complex system interconnectedness - but again what the multiple ageing-related changes in DNA methylation represent in blood (cell type changes etc.)

3. The statement in Abstract and elsewhere that “we locate the most important CpGs'” ignores the fact that they limit their analysis to only the 353 CpG from the total DNA methylome of 28 million CpGs to begin with. As Horvath has stated there is no evidence that the CpGs in the Horvath clock are especially functional over and above many other CpGs and reasonable clocks can be constructed from even a random selection of CpGs - there are abundant potential CpGs that can be exploited in clocks [3]. The statement “largest in influence" and “which may also play a crucial role in the process of ageing" (Introduction, Line 94) again implies these small fraction of 353 CpGs are uniquely special [4].

We have refined the mentioned statements, that refer specifically to the studied subset of CpGs in the revised version and put a caveat to the end of the Introduction to remind the reader that the analysis should be extended to get relevant results. (Ref.[4] from the referee report has been also incorporated into the manuscript, as described in the answer to Major point no. 9.)

• Good to acknowledge this caveat to this analysis.

4. Age-related change in DNA methylome is in fact widespread with up to 15-30% of all CpG sites in the genome associated with age-related changes and these are not all called `clock CpGs' (Introduction line 18). Change can be random fashion due to epigenomic drift [5], directional, or show increased variability with age[6]. Also, the statement regarding the directionality of “clock CpGs that are hypermethylated" (Introduction line 35) is an oversimpli_cation. Teschendorf et al. identified an enrichment in an early promoter-focused array for age-related CpGs that were hypermethylating in the Targets of Polycomb Target gene promoters, but genome-wide hypomethylation predominates. Both hypo- and hypermethylated loci contribute to the various published clocks.

We have rephrased the part of the text introducing the clock CpGs, now mentioning that age related CpGs are actually quite common, and that not all of them are called as clock CpGs. The revised version of the manuscript is now citing Refs[5,6] from the referee report. We also replaced 'hypermethylation' by 'age related change' in the sentence referring to the work by Teschendorf et al.

• The term ‘clock CpGs’ implies that these specific CpGs possess special properties. Therefore, this term only leads to confusion and would be better to remove – there are multiple CpGs in the DNA methylome that could be included, or not, into differently constructed DNA methylation clocks [Field et al].

5. The statement in the Introduction that there are “connections between the CpGs themselves” (line 75) is as expected. Clearly all well-known ageing effects lead to co-ordinated changes across the entire DNA methylome these include those driven by cell-type specific epigenomics where changes in cell proportion will led to variation (including the age-related myeloid skew [7], T cell exhaustion) [8], polycomb target hypermethylation [9], bivalent domain hypermethylation [10], etc. These known systemic effects will be seen as networks of

age-related change.

We are especially grateful for this comment, providing extra support for the networked approach we use to study DNA methylation and ageing. This is now incorporated into the text (together with the references), however at a somewhat earlier point, where we first mention connections between the CpGs.

• Good to now include this information.

6. Distinct biological processes drive the observed age-related hypermethylation and hypomethylation. Furthermore, the baseline DNA methylation state is strongly driven by genetics being highly CpG density dependent [11].

We included this important point (together with the reference) in the revised version where we list the diffculties of constructing multi-tissue DNA methylation-based age estimators.

• Good to now include this information.

7. The statement (line 53) that “we cannot really point out any of these CpGs as being more important than others" is as completely expected in the way that the elastic net regression Horvath clock was designed. CpGs were selected not for their individual strength but chosen for their power to work collectively to parsimoniously capture ageing over the lifecourse. In fact, this is clearly demonstrated by the fact the strongest and most robust individual CpG pan-tissue changes from the ELOVL2 locus [12,13] were not included in the clock. Additionally, an accurate clock has been devised using just 3 CpGs [14].

We agree that this is statement is somewhat evident, nevertheless we would like to keep it in the Introduction for helping non-expert readers in understanding the basis of our study. The sentence before this statement already mentioned that the correlation between age and the methylation of individual CpGs from Horvath's clock is weak; we have rephrased this sentence based on this comment, now citing Refs[12,13] from the referee report. Ref.[14] from the referee report was already cited in the original manuscript as Ref.[51] in the Discussion.

• However, the authors should include at this point why the specific methodology (elastic net) employed in the construction of the Horvath clock would contribute to this observation.

8. The discussion of “control properties" of CpGs is consistent with the Elastic Net picking those CpGs that work well together. Thus, the results regarding network identification and properties have ignored this and the limited CpGs this has been exacted from e.g. Results (line 112). Why were not all the 850,000 CpGs from the EPIC array analysed in the network analysis rather than just 353? Conclusion statements regarding how a “network approach can bring new insight into methylation-related studies, providing a very interesting direction

for further research" (Line 389) are clearly limited when restricted to only these 353 CpGs and known biology not taken into account.

Analysing 850k (new EPIC array) or even 27k CpGs (older methylation array) is unfortunately not feasible computationally, due to the combinatorial explosion of the all-to-all nature of our analysis. This was the main reason why we have used only this limited set. In the updated version we call the readers' attention to this limitation. The network we analysed can be viewed as a small sub-graph from the several orders of magnitude larger system of the whole methylome. A relevant related question is how do the interesting hierarchical and control properties we observed change when we scale up the network size? During the review process as a first step we have repeated our analysis on a network roughly 10 times larger obtained as follows. We took the 353 CpG dinucleotides in Horvath's clock one by one as a response variable, and carried Lasso regressions on the whole 450K CpG array appearing in the input data, where we marked the regressors (CpGs) obtaining a non-zero coefficient at least once. These marked CpGs along with the 353 CpGs in Horvath's clock defined an extended set of nodes, counting altogether 2036 CpGs. Among this larger set of nodes, the links were obtained based on LassoCV regression, following the network construction method described in the paper. We thresholded the links based on the absolute value of the regression coefficients to ensure that the average degree of the extended network becomes the same as in case of the original network studied in the paper. The results of the hierarchy analysis on this extend network are shown in Fig.1. As we can see, this network is again significantly more hierarchical compared to its random configuration model counterparts, similarly to the original network studied in the paper. Furthermore, the outcome of the control centrality analysis, shown in Fig.2., was also resembling to results we obtained for the network based solely on Horvath's clock.

• It would be good to demonstrate these in others clocks such as Hannan et al, GrimAge, SkinBlood etc to reinforce the biology.

9. The authors need to explain and understand more precisely what the concept of `biological age' and predicators of this represent [15]. The initial Horvath clock was devised as an attempt at a `pan-tissue' clock (which it was highly successful in although caveats remain [16,17]). It is in fact a `composite' clock [3] capturing both forensic and biological age but neither perfectly. The authors need to understand and integrate the current knowledge and issues regarding DNA methylation clocks - as discussed recently by the epigenomics community [4].

We revised the part in the Introduction mentioning the 'biological age' according to Refs.[3,15] in the referee report, which are now also cited in the manuscript. In addition, beside the success of Horvath's clock, we now mention the existence of related caveats together with citing Refs[16,17] from the referee report. Finally, key challenges and issues discussed in Ref.[4] of the referee report are also listed in the revised version (together with a citation to the paper).

• Good to acknowledge and expand on this important point.

10. The statements regarding “Modifying the predicted age by perturbing the methylation network” need to be put in the context that they are interpreting a `biomarker' of biological ageing.

We have checked that we always refer to the adjustment of the "estimated" or "predicted" age and not true biological age. As indicated in the answers for other questions, we have put caveats concerning the interpretation both into the Introduction and Discussion.

• Good to include these caveats.

11. Unclear what “more aligned with the 'natural direction of ageing'." (Line 283) means biologically?

The methylation values can be considered as coordinates of a multidimensional vector space. E.g. if we consider the 353 CpGs it will be a 353 dimensional space. Each patient's methylation measurement is a point in this space. Since methylation values are not random, the points do not cover the whole space, rather they are constrained to a (potentially curved) subspace. Projection techniques like the linear PCA or the recently popular non-linear t-SNE can reveal the most extended directions and are widely used to visualise the most important features of a high-dimensional data set. The principal directions can often be interpreted as biological features. For example, the regression techniques used for age estimation identify such linear subspace. Changing few methylation values would move points according to the vector span by the linear combination of the corresponding axes, but the resulting position of the point may not necessarily stay on the "biologically allowed" subspace. As methylation values are part of an interacting network, change of one value cannot happen in isolation. In this part of the paper we describe this and show that by taking into account the cascading changes on our control network lead to changes that keep the points on the "biologically allowed" subspace in contrast to isolated (without following control cascades) changes that move points away from the subspace.

• The construction by elastic net will accentuation this interconnectedness.

12. In the Discussion the statement `Horvath's clock is showing non-trivial hierarchical and control properties' how is this unexpected? Furthermore, how would that be different from a random selection of array-derived CpG probes?

In this study we represent the system of CpG dinucleotides as a network, and although we do not expect this to behave as e.g., an Erdos-Renyi random graph, still, the non-trivial nature of the interrelations can in principle be manifested in several different ways. E.g., a network can be different from a random graph in terms of its degree distribution, can display a community structure (that is absent in random graphs), may show assortativity or disassortativity, etc. In our view, it is not straightforward that a network ought to have a hierarchic structure (accompanied by interesting control properties) just because it represents biological data. When considering a random baseline for comparison, we have to take into account that hierarchy measures are quite sensitive to the overall link density in networks. Based on that, we have chosen the configuration network ensemble to serve as the baseline, where the random graphs correspond to uniformly drawn samples from all possible graphs with the same degree sequence as the original network, as mentioned in the Results section related to Fig.2. In this way we cancel out any possible uncertainty in the GRC coming from either a change in

the overall link density or from a difference in the degree distribution. Selecting random CpG probes is a very interesting idea, however, we would leave this to be the subject of further study, where also the size of the examined network might be increased (the first preliminary results of this analysis are described in the answer to Major comment no.8). Nevertheless, based on the results we have seen for the network of Horvath's clock, we expect both the entire network between all CpGs and randomly chosen sub-graphs from this to display hierarchical properties.

• A randomisation as well as analysis of the other clocks would help consolidate interpretation.

13. The statements regarding the functional implications of individual CpGs in the Discussion need to be more clearly caveated [8].

The description of the biological function of the genes was moved to the appendix (also because another referee found this part too long) and caveats were added.

• Good

14. In the Conclusion (line 374) the statement “substantially more hierarchical compared to a random Graph" does not take into consideration the biological nature of these data.

The concept of hierarchy in this work was introduced from a network theoretic point of view, e.g., the hierarchy measure we apply was used in social and technological networks as well in the literature. The random graph ensemble serving as a baseline preserves the degree distribution of the original network, thus, the most fundamental component of the network structure is not affected by the randomisation. In this light, the observation of a significantly higher GRC value in the original network compared to the random ensemble is already interesting from a pure network theoretic point of view. Nevertheless, we believe that this can be interesting for biologists as well, as it shows a non-trivial wiring between the CpG dinucleotides, where we can reach the majority of the network from nodes at the top of the hierarchy in just a few steps, whereas we cannot

from bottom nodes.

• The authors need to appreciate what these biological data represent to help explain why a hierarchical structure is observed.

Minor

• Good - all minor points have been corrected

Reviewer #2: What I concerned most is the biological significance and stability of the age-related methylation hierarch relationships as described in this study. However, the authors still focused only on the 353 CpGs involved in Horvath’s clock, which is far from sufficient to represent the age-related CpG sites. More seriously, the authors still focused on and analyzed only one methylation dataset, this is far from enough to reach the conclusion: ‘the methylation hierarch relationships really happen during ageing’. Although the authors stated that ‘Analysing 850k (new EPIC array) or even 27k CpGs (older methylation array) is unfortunately not feasible computationally, due to the combinatorial explosion of the all-to-all nature of our analysis’, this explanation is unacceptable for me. In any case, more independent methylation dataset (e.g., 450K) to replicate their results is indispensable. Overall, the stability of the findings is doubtful which largely devalues the work.

**Have all data underlying the figures and results presented in the manuscript been provided?**

Reviewer #1: Yes

Reviewer #2: Yes

PLOS authors have the option to publish the peer review history of their article (what does this mean?). If published, this will include your full peer review and any attached files.

Reviewer #1: No

Reviewer #2: No
---

## [Decision Letter · Decision Letter 2]

20 Apr 2021

Dear Dr. Pollner,

Thank you very much for submitting your manuscript "Hierarchy and control of ageing-related methylation networks" for consideration at PLOS Computational Biology.

As with all papers reviewed by the journal, your manuscript was reviewed by members of the editorial board and by several independent reviewers. In light of the reviews (below this email), we would like to invite the resubmission of a significantly-revised version that takes into account the reviewers' comments.

In particular, Reviewer 2 again voiced substantial and very important critiques that must be addressed to this Reviewer's satisfaction, in order for your paper to be processed further.

We cannot make any decision about publication until we have seen the revised manuscript and your response to the reviewers' comments. Your revised manuscript is also likely to be sent to reviewers for further evaluation.

Sincerely,

Ilya Ioshikhes

Deputy Editor

PLOS Computational Biology

Douglas Lauffenburger

Deputy Editor

PLOS Computational Biology

Reviewer's Responses to Questions

**Comments to the Authors:**

Reviewer #1: Please see attachment

---

- Pollner et al. have now provided a second round of responses to reviewers’ queries, as below.

- The authors have, commendably, performed extensive additional work, including further analysis of other epigenetic ageing clocks, random CpGs, and they have tempered statements regarding causation of these clock-related DNA methylation changes.

Response to Reviewer 1:

1. We hope that the amendments made in the previous and current version regarding points 1., 2., and 5., related to the relevant biological processes connected to methylation profile changes are now satisfactory.

2. We have made further changes to this part to avoid the possible interpretation that methylation level changes are the cause of ageing-related changes. In the current version we mention first the changes in cell proportion, including the age-related myeloid skew, T cell exhaustion, polycomb target hypermethylation, bivalent domain hypermethylation, which lead to coordinated modifications of the entire methylome, that in turn can be also interpreted as a network of age-related change in the methylation levels of the CpGs.

3. We thank the referee for accepting our response.

4. We have removed the term 'clock CpG' entirely from the paper.

5. We thank the referee for accepting our response.

6. We thank the referee for accepting our response.

7. In the new version we now explicitly mention when introducing Eq.(1) that Horvath's clock is based on elastic net regression. Furthermore, after the sentence referred in this point we inserted a short description of the parameter selection in the elastic net approach.

8. We have applied our framework to the Skin-Blood clock and Hannum's clock, receiving very similar results as already shown for Horvath's clock. The corresponding Figures have been placed into the Supporting Information (Sect.S2, Figs.S2-S7.) now accompanying the paper. The analysis for both of these further epigenetic clocks has shown that the methylation network composed of their CpGs is hierarchical, where the control centrality of the nodes is in positive correlation with the position of the nodes in the hierarchy. In addition, the chance to achieve a larger expected change in the estimated age when perturbing the methylation levels seemed to be higher for nodes close to the top of the hierarchy with large control centrality. These results are in very clear analogy with the results we discuss for Horvath's clock in the main text.

9. We thank the referee for accepting our response.

10. We thank the referee for accepting our response.

11. We have added a short reminder for the readers about the fact that CpGs in Horvath's clock were selected using elastic net regression.

12. Beside the analysis of the Skin-Blood clock and Hannum's clock we have also studied methylation networks composed of randomly chosen CpGs with a fixed size equal to that of Horvath's clock (353 nodes). The corresponding results are presented in the Supporting Information (Sect.S3, Figs.S8-S9.), indicating that these networks display quite similar properties compared to the previously studied networks representing epigenetic clocks. On the one hand, the hierarchy measure (the GRC) in their link randomised counterparts is on average lower compared to the GRC value of the original network structure encoding the inferred relations between the methylation levels. On the other hand, the control centrality of the nodes is in positive correlation with their position in the hierarchy. When comparing the GRC value obtained for Horvath's clock with the GRC distribution of the random methylation network we can observe that the hierarchy measure for Horvath's clock is above the average at all studied m-parameters. However, its value is not an outlier, in the units of the standard deviation _ of the random distribution the difference is roughly between 1 � and 2 �, depending on m. Thus, the methylation network of Horvath's clock is resembling a methylation network with random CpGs where the hierarchy of the system is somewhat larger than the average, but not outstandingly large. By putting together the results obtained for networks representing epigenetic clocks and for the networks based on CpGs chosen uniformly at random from the 450k methylation array, we can conclude that basically any methylation network constructed according to our framework can be expected to display a hierarchical structure accompanied by control centrality values in positive correlation with the node position in the hierarchy. A remaining question of interest whether the hierarchy rankings obtained for small networks have an indicative value for the importance of the nodes we would observe in larger methylation networks where the size of set of CpGs taken into consideration is extended. Relating to that we have also examined mixed networks, where 10% of the CpGs were from the top of the hierarchy of Horvath's clock, and the rest of the nodes were chosen at random. The results (Fig.8. in the new version of the submission) show that hierarchy positions are conserved to a considerable extent across the different networks. This is promising for possible future research where the structure of larger parts from the methylome may be studied in small fragments analysed in a parallel fashion.

13. We thank the referee for accepting our response.

14. In regard to this point, the new version of the manuscript now mentions the relevant ageing related effects that are known to lead to coordinated methylation changes at this part of the Discussion.

- All my concerns have been sufficiently answered

Reviewer #2: In the new revision, the authors attempted to confirm the stability of the hierarchy of ageing-related networks by considering the CpG sites in Skin-blood clock and Hannum’s clock. However, these analyses are still performed by using only one Human Methylation 450K dataset (GSE40279) even the reviewer has already pointed out last time. This strategy does not support the view that the finding for the CpG sites of concern is general. As the authors explained -- “Analysing 850k (new EPIC array) or even 27k CpGs (older methylation array) is unfortunately not feasible computationally, due to the combinatorial explosion of the all-to-all nature of our analysis” or “Scaling up the network size in our analysis to the level of the whole 450k array would take more than 150 years; thus, it is truly not feasible”, but it means that the analysis of CpG sites in ‘Epigenetic Clock’ (e.g., Horvath’s clock) in another independent methylation dataset (e.g., GSE55763 that contains over 2600 samples) is theoretically possible and worthy, even perhaps I underestimated the difficulty. Overall, I have to say the current revision is still inconclusive.

**Have the authors made all data and (if applicable) computational code underlying the findings in their manuscript fully available?**

Reviewer #2: None

PLOS authors have the option to publish the peer review history of their article (what does this mean?). If published, this will include your full peer review and any attached files.

Reviewer #1: No

Reviewer #2: No

**Have all data underlying the figures and results presented in the manuscript been provided?**

Reviewer #1: Yes
---

## [Decision Letter · Decision Letter 3]

5 Aug 2021

Dear Dr. Pollner,

We are pleased to inform you that your manuscript 'Hierarchy and control of ageing-related methylation networks' has been provisionally accepted for publication in PLOS Computational Biology.

Best regards,

Ilya Ioshikhes

Deputy Editor

PLOS Computational Biology

Douglas Lauffenburger

Deputy Editor

PLOS Computational Biology

Reviewer's Responses to Questions

**Comments to the Authors:**

Reviewer #2: In the revised manuscript, the authors have explored the findings in an independent methylation dataset. They described the differences of the results between the two datasets and discussed the possible reasons. Overall, the study has some scientific significance, although the evidence is still not very strong. In this regard, the authors shall lower down their tone on some statements. I have no further comments.

**Have the authors made all data and (if applicable) computational code underlying the findings in their manuscript fully available?**

Reviewer #2: None

PLOS authors have the option to publish the peer review history of their article (what does this mean?). If published, this will include your full peer review and any attached files.

Reviewer #2: No

---

## [Editor Report · Acceptance letter]

3 Sep 2021

PCOMPBIOL-D-20-00368R3 

Hierarchy and control of ageing-related methylation networks

Dear Dr Pollner,

I am pleased to inform you that your manuscript has been formally accepted for publication in PLOS Computational Biology. Your manuscript is now with our production department and you will be notified of the publication date in due course.

With kind regards,

Andrea Szabo
